# Ceramic thin-film composite membranes with tunable subnanometer pores for molecular sieving

Xuechen Zhou[1], Rahul Shevate[2], Dahong Huang[1], Tianchi Cao[1], Xin Shen[1], Shu Hu [1], Anil U. Mane[2], Jeffrey W. Elam [2], Jae-Hong Kim [1] ✉ & Menachem Elimelech [1] ✉

Ceramic membranes are a promising alternative to polymeric membranes for selective separations, given their ability to operate under harsh chemical conditions. However, current fabrication technologies fail to construct ceramic membranes suitable for selective molecular separations. Herein, we demonstrate a molecular-level design of ceramic thin-film composite membranes with tunable subnanometer pores for precise molecular sieving. Through burning off the distributed carbonaceous species of varied dimensions within hybrid aluminum oxide films, we created membranes with tunable molecular sieving. Specifically, the membranes created with methanol showed exceptional selectivity toward monovalent and divalent salts. We attribute this observed selectivity to the dehydration of the large divalent ions within the subnanometer pores. As a comparison, smaller monovalent ions can rapidly permeate with an intact hydration shell. Lastly, the flux of neutral solutes through each fabricated aluminum oxide membrane was measured for the demonstration of tunable separation capability. Overall, our work provides the scientific basis for the design of ceramic membranes with subnanometer pores for molecular sieving using atomic layer deposition.

The deployment of clean energy technologies is boosting the global metal demand to an unprecedented extent[1,2]. For instance, the annual lithium demand was projected to be 2.5 million tons in 2030, 8.7 times the demand in 2020[3]. With multiple metal demands approaching their global reserves (e.g., 14 million tons for lithium[4]), there is a heightened interest in augmenting the supply of critical metals by either recycling precious metals from industrial waste streams or harvesting metals from unconventional sources such as seawater[5,6]. Recovering valuable metals (e.g., nickel and copper) from waste streams also helps lower the toxicity of the streams and offsets the cost of waste treatment[7,8]. Given the complexity of waste streams, there is a crucial need to develop highly precise separation processes for metal recovery.

Membrane technology has been widely used for selective separations due to its great separation capability[9,10]. Designing polymeric membranes with functional groups favorably interacting with desired ionic species has been demonstrated as an efficient pathway to selectively transport these ions from the feed streams[5,11]. The attraction between membrane pores and ions compensates the energy penalty for ion dehydration at the pore entrance and thus favors their passage[12]. Subnanometer membrane pores are preferred so that the non-selective bulk-phase ion permeation can be minimized[11]. Despite the versatility of the fabrication strategy and the tunability of the membrane chemistry, one major concern regarding employing polymeric membranes for selective separation is their instability under harsh operating conditions

[1]Department of Chemical and Environmental Engineering, Yale University, New Haven, CT, USA. [2]Applied Materials Division, Argonne National Laboratory, Lemont, IL, USA. ✉e-mail: jaehong.kim@yale.edu; menachem.elimelech@yale.edu

(e.g., high or low pH, presence of oxidants, and high temperatures). For instance, polyamide thin-film composite membranes, the state-of-the-art reverse osmosis membranes, are vulnerable to chlorine attack, substantially compromising their separation performance[13–15].

To further enhance the reliability and operational feasibility of the membrane separation process, it is imperative to design selective membranes with improved stability. One promising alternative is using ceramic membranes (e.g., aluminum oxide[16], zirconium oxide[17], and titanium oxide[18]) for selective separation, given their tolerance for harsh chemical conditions such as extreme solution pH and chlorine cleaning[19,20]. Nevertheless, current fabrication techniques, mainly the sol-gel method[21,22], rely on the assembly of particulates to create the separating layers and thus can only construct microfiltration or ultrafiltration ceramic membranes[23] (pore size >2 nm). Therefore, to impart precise molecular sieving for ceramic membranes, especially for ion–ion separation, it is crucial to develop ceramic membranes with subnanometer pores.

A molecular-level design of the active layer is required to construct ceramic membranes with subnanometer pores. One proposed strategy is calcinating the metal-organic hybrid films prepared using the molecular layer deposition (MLD) technique[24–26]. Specifically, in these MLD processes, metal atoms and organic linkers are deposited on the substrate surface, layer by layer, by sequentially exposing the substrates to the metal-organic precursors (e.g., titanium tetrachloride) and the bifunctional co-reactants (mainly ethylene glycol, EG). By burning off the organic linkers within the hybrid films, (sub) nanometer pores were supposed to be generated. Despite the attempted molecular-level design, the fabricated ceramic membranes showed limited capability in differentiating between the transport of different ions[26]. This inconsistency is attributable to the condensation of EG during the MLD deposition, creating large defects after calcination[27]. Hence, to realize the application of this sacrificial template strategy in constructing defect-free ceramic membranes with subnanometer pores, it is imperative to build hybrid films with distributed molecular-scale sacrificial segments. However, this objective can barely be realized with the conventional MLD process as EG is among the most volatile bifunctional co-reactants.

In this work, we use monofunctional-alcohol-modulated atomic layer deposition (ALD) to construct aluminum oxide (AlO$_x$) membranes with tunable subnanometer pores. Taking the methanol-modulated ALD deposition as an example, we first demonstrate the proposed deposition strategy by monitoring the generation and incorporation of methoxy groups within the deposited films. With the optimized deposition and calcination conditions, we then fabricate AlO$_x$ membranes with continuous subnanometer pores by burning off the incorporated methoxy groups. The selectivity of these membranes toward the transport of monovalent and divalent salts is tested. Lastly, permeation of neutral solutes through the AlO$_x$ membranes prepared with the modulation of different alcohols (i.e., methanol, ethanol, and isopropanol) is characterized to highlight the tuning of membrane pore size and their separation capability.

## Results
### Methanol-modulated atomic layer deposition of aluminum oxide

A methanol (CH$_3$OH)-modulated ALD process was used for precise control of the pore structure within the designed ceramic membranes (Fig. 1). In this ALD process, a methoxy group (−OCH$_3$)-decorated

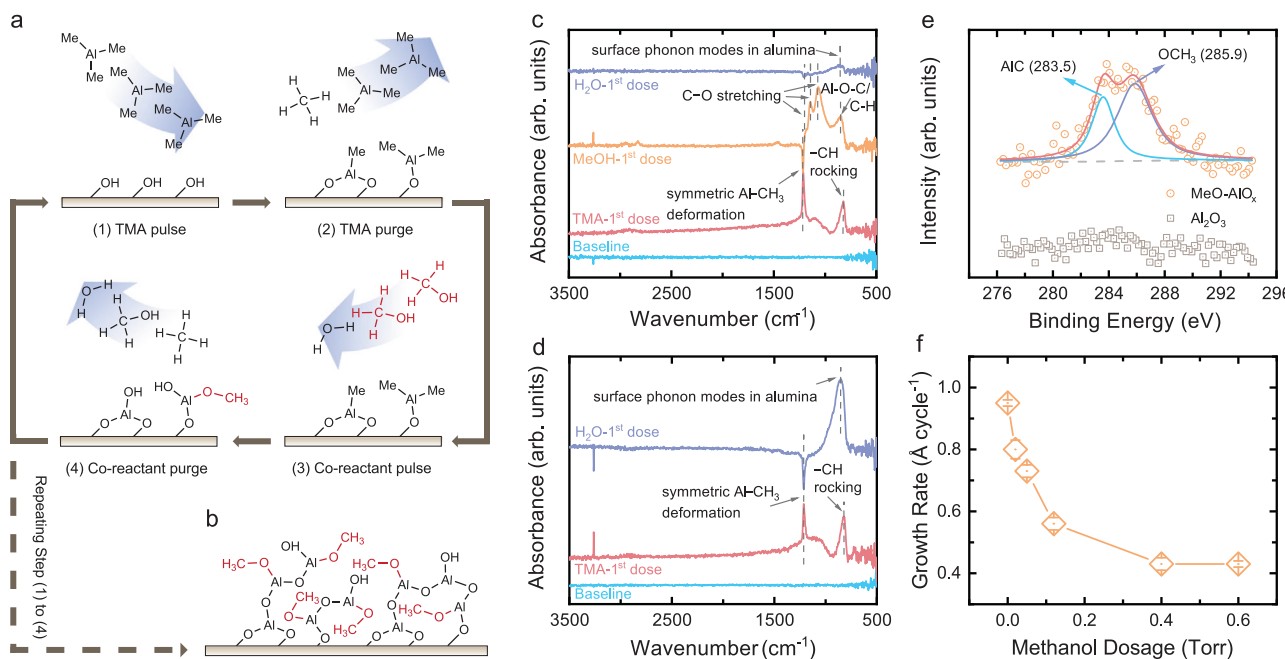

**Fig. 1 | Fabrication of the MeO-AlO$_x$ film using the CH$_3$OH-modulated ALD process. a** Schematic diagram describing one ALD cycle of MeO-AlO$_x$ deposition, which consists of four steps: (1) TMA pulse; (2) TMA purge; (3) co-reactant pulse (i.e., a mixture of CH$_3$OH and H$_2$O); and (4) Co-reactant purge. Only a single atomic layer of MeO-AlO$_x$ is deposited during one ALD cycle. **b** Schematic diagram illustrating the chemical structure of the deposited MeO-AlO$_x$ film, where −OCH$_3$ groups (highlighted in red) are incorporated into the Al-O framework during the deposition. **c** Differential FTIR spectra recorded during the CH$_3$OH-modulated ALD deposition after one pulse of TMA (red), CH$_3$OH (orange), and H$_2$O (purple). The original FTIR spectra after each pulse can be found in Supplementary Fig. 1a. **d** Differential FTIR spectra recorded during the standard Al$_2$O$_3$ ALD deposition

after one pulse of TMA (red) and H$_2$O (purple). The original FTIR spectra after each pulse can be found in Supplementary Fig. 1b. **e** C 1s XPS spectra of the ALD deposited Al$_2$O$_3$ film and the CH$_3$OH-modulated ALD deposited MeO-AlO$_x$ film. Argon sputtering was conducted to remove the adventitious carbon contamination before the XPS test. **f** Dependence of the MeO-AlO$_x$ film growth rate on the CH$_3$OH dosage during the ALD deposition. For each CH$_3$OH dosage, the TMA and H$_2$O dosages were fixed at 0.25 and 0.06 Torr, respectively. The growth rate was characterized by measuring the pore narrowing rate when conducting the ALD deposition on top of AAO substrates (Supplementary Figs. 4 and 5). Error bars represent standard deviations from the linear fitting.

aluminum oxide film (MeO-AlO$_x$) can be deposited on the substrate surface by sequentially exposing the substrates to three different chemicals: trimethylaluminum (Al(CH$_3$)$_3$, TMA), H$_2$O, and CH$_3$OH (Fig. 1a, b). Compared to EG (HOCH$_2$CH$_2$OH) used in the MLD process in previous studies, both H$_2$O and CH$_3$OH are more volatile (boiling points of 100, 64.7, and 198 °C for H$_2$O, CH$_3$OH, and HOCH$_2$CH$_2$OH, respectively), reducing the condensation of co-reactants during the deposition (at 150 °C) and the formation of defects after the calcination step[27].

Figure 1a illustrates a typical cycle of the proposed ALD deposition. In the first half-cycle, hydroxyl groups (−OH) on the substrate surface react with the dosed TMA precursors and are converted to methyl aluminum groups (−AlCH$_3$). Due to the self-limiting nature of the reaction, only a single layer of −AlCH$_3$ groups can be deposited at each cycle. After purging the reaction chamber with argon gas, a mixture of CH$_3$OH and H$_2$O was dosed sequentially without a purge break, with both co-reactants reacting with the −CH$_3$ groups in the following second half-cycle. Specifically, −OH groups are regenerated through the reaction between the H$_2$O molecules and the −CH$_3$ groups, which can act as the active sites for the next cycle of ALD growth. Concomitantly, the remaining −CH$_3$ groups are converted to −OCH$_3$ groups after reacting with CH$_3$OH; these −OCH$_3$ groups inhibit TMA binding in the next ALD cycle. The two sequential reactions in each ALD cycle can be described as follows:

$$AlOH^* + Al(CH_3)_3 \rightarrow AlOAl(CH_3)_2{}^* + CH_4 \qquad (1)$$

$$AlOAl(CH_3)_2{}^* + H_2O + CH_3OH \rightarrow AlOAl(OCH_3)OH^* + 2CH_4 \qquad (2)$$

where the asterisks denote surface species.

In situ Fourier transform infrared (FTIR) spectroscopy was carried out to probe the surface reactions during the proposed ALD deposition, and the surface chemistry was examined after each step of precursor dosage. To explicitly demonstrate the surface change after exposure to the mixture of CH$_3$OH and H$_2$O, FTIR spectra were obtained separately after sequentially exposing the substrate to CH$_3$OH and H$_2$O (Supplementary Fig. 1). Differential FTIR spectra were obtained by subtracting the spectra from the corresponding ones acquired in the previous step to highlight the changes occurring during each step (Fig. 1c, d). Positive-going peaks suggest the generation of new species, while negative-going features indicate the consumption of surface functional groups.

According to Fig. 1c, two positive peaks with wavenumbers of 821 and 1209 cm$^{-1}$ appear after the dose of TMA. The peak at 821 cm$^{-1}$ is the characteristic of −CH rocking[28], and the one at 1209 cm$^{-1}$ represents the symmetric Al−CH$_3$ deformation[29]. Both peaks support the generation of −AlCH$_3$ groups on the substrate surface after reacting with TMA. Following the dose of CH$_3$OH, a negative peak at 1209 cm$^{-1}$ was observed, indicating the consumption of surface −AlCH$_3$ groups. Concomitantly, positive peaks with wavenumbers of 1205, 1145, and 1068 cm$^{-1}$ appear, which can be assigned to C−O stretching[30]. The appearance of C−O groups is consistent with the anchoring of −OCH$_3$ groups (Fig. 1a). The positive peak appearing at 848 cm$^{-1}$ can be assigned to Al−O−C, affirming the formation of Al−OCH$_3$ species. Finally, after the H$_2$O dosage, the negative features at 1205, 1145, and 1068 cm$^{-1}$ indicate the partial consumption of −OCH$_3$ groups whereas the broad positive peak between 768 and 1077 cm$^{-1}$ represents the surface phonon modes in alumina[30]. The observed spectrum change suggests that the anchored −OCH$_3$ groups are vulnerable to H$_2$O attack and can be substituted with −OH groups:

$$AlOCH_3{}^* + H_2O \rightarrow AlOH^* + CH_3OH \qquad (3)$$

As a control, when we only provided H$_2$O after the TMA exposure (i.e., a conventional ALD deposition of Al$_2$O$_3$; Supplementary Fig. 2), a significantly higher positive peak representing the surface phonon modes in alumina was observed between 768 and 1077 cm$^{-1}$ (Fig. 1d), consistent with the fact that adding CH$_3$OH as a co-reactant inhibits the growth of the alumina film.

To further confirm the incorporation of −OCH$_3$ groups within the deposited MeO-AlO$_x$ film despite the substitution with −OH groups when using the mixture of H$_2$O and CH$_3$OH as the co-reactant (Eq. 3), we characterized the chemical composition of the deposited film using X-ray photoelectron spectroscopy (XPS). In situ argon ion sputtering was applied before the XPS measurement to clean adventitious carbon so that the carbon content within the deposited films can be explicitly detected. According to Fig. 1e, carbon species were found within the deposited MeO-AlO$_x$ film, whereas no significant peak was observed within the C 1s XPS spectrum of the Al$_2$O$_3$ film deposited in the absence of CH$_3$OH. Additional deconvolution of the C 1s peak reveals that the detected carbon species within the MeO-AlO$_x$ film has two components. Specifically, the peak at 285.9 eV is consistent with previously reported oxygen-bound carbon within −OCH$_3$ groups[31], and the other peak with binding energy of 283.5 eV can be ascribed to the metal carbide (Al−C here) which was generated during the argon sputtering process[32].

To demonstrate the tuning of −OCH$_3$ group density within the deposited films, the film growth rates at various CH$_3$OH dosages were characterized (Supplementary Figs. 4 and 5). As shown in Fig. 1f, when H$_2$O was used as the only co-reactant (i.e., 0 Torr of CH$_3$OH dosage), the growth rate of the film was 0.95 Å cycle$^{-1}$, in agreement with previously reported Al$_2$O$_3$ ALD film growth rate at 150 °C[33,34]. As the CH$_3$OH dosage increases during the ALD deposition, the film growth rate decreases continuously, suggesting that more −CH$_3$ groups are converted to −OCH$_3$ groups (terminators) rather than −OH groups (active sites) during the co-reactant exposure. When the CH$_3$OH dosage is higher than 0.40 Torr, the film growth rate is almost constant (~0.40 Å cycle$^{-1}$), likely due to the near saturation of the substrate surface with −OCH$_3$ groups[35].

**Fabrication of aluminum oxide thin-film composite membranes**
As illustrated in Fig. 2a, an anodic aluminum oxide (AAO) membrane was used as the substrate for membrane fabrication, and a continuous MeO−AlO$_x$ film was constructed on top of the substrate using the proposed ALD process. The ALD coating first narrows the pores of the AAO substrate and then forms a continuous layer[36]. The minimum number of ALD cycles required to completely block the AAO pores was calculated based on the measured pore radius (~15.4 ± 0.7 nm, Supplementary Fig. 5) and the film growth rate (depending on the precursor dosage, Fig. 1f). Scanning electron microscope (SEM) was used to verify the construction of the continuous MeO-AlO$_x$ layer; compared to the pristine AAO substrate (Fig. 2b), no microscopically visible pore structure was observed on top after the ALD coating (Supplementary Fig. 6a). A cross-sectional SEM image shows that the ALD coating only spans the top ~100 nm of the AAO substrate (Fig. 2c), suggesting a thin-film composite (TFC) structure of the constructed membranes. This observation agrees well with the small TMA dose time (0.07 s) we employed for deposition, which is significantly lower than that required for the AAO pore saturation (calculated saturation dose time: 14.7 s; Supplementary Note 1). In addition, the progressive decrease in pore diameter with increasing ALD cycles adds significant resistance to TMA diffusion inside the AAO pores, resulting in the formation of the TFC structure on AAO substrates[37].

The fabricated composite membrane was then calcinated to burn off the −OCH$_3$ groups to create pathways to both water and solute transport (Fig. 2a, denoted as AlO$_x$ membranes). No visible destruction of the ALD layer was observed after being calcinated for 5 h under an air atmosphere (Fig. 2d), and the membranes became hydrophilic

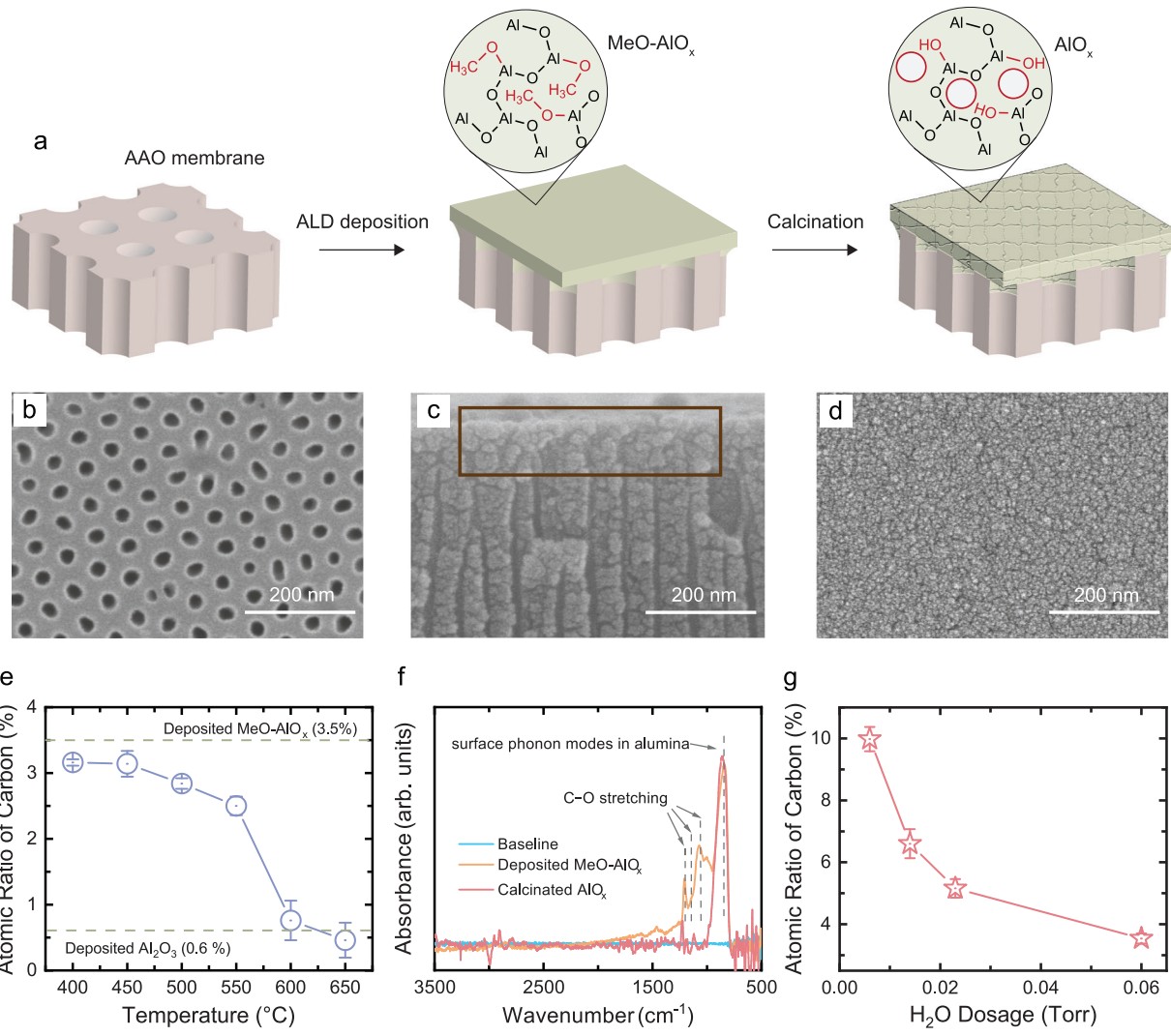

**Fig. 2 | Construction of the AlO$_x$ TFC membranes with subnanometer pores.**
**a** Schematic diagram showing the fabrication of the AlO$_x$ TFC membranes. Briefly, a continuous MeO-AlO$_x$ film was deposited on top of the AAO substrate using ALD. This composite membrane was then calcinated to burn off the −OCH$_3$ groups to create continuous pathways for water and solute permeation. **b** SEM image of the top surface of the pristine AAO substrate. **c** SEM image of a cross-section of the MeO-AlO$_x$ TFC membrane. A continuous integral layer (highlighted in the brown box) was observed near the top of the AAO substrate. **d** SEM image of the top surface of the AlO$_x$ TFC membrane. Scale bars are 200 nm. **e** Dependence of the carbon atomic ratio within the ALD deposited MeO-AlO$_x$ film on the calcination temperature. All the carbon species are burnt off at calcination temperatures higher than 600 °C. Error bars represent standard deviations of 5 measurements. **f** FTIR spectra of the ALD deposited MeO-AlO$_x$ film and the AlO$_x$ film after being calcinated at 600 °C. **g** The impact of H$_2$O dosage during the ALD deposition on the atomic ratio of carbon within the deposited MeO-AlO$_x$ ALD layers. For each H$_2$O dosage, the TMA and CH$_3$OH dosages were fixed at 0.25 and 0.40 Torr, respectively. Error bars represent standard deviations of 5 measurements.

(Supplementary Fig. 6b). The calcination temperature was optimized to ensure the complete removal of the −OCH$_3$ groups. Briefly, the deposited MeO-AlO$_x$ films were calcinated at different temperatures and their chemical composition after the calcination, especially the atomic carbon ratio, was characterized using the XPS technique. As shown in Fig. 2e, carbon was found to make up 3.5% of the deposited film, and this ratio was maintained nearly constant when the film was calcinated at a temperature lower than 450 °C. Raising the calcination temperature above 450 °C resulted in a continuous decline of the atomic ratio of carbon within the film. When the calcination temperature was higher than 600 °C, no residual carbon was observed (Supplementary Fig. 7), indicating that most of the −OCH$_3$ groups within the deposited ALD layer were burned off. The removal of the −OCH$_3$ groups after the calcination was further examined using FTIR. According to the FTIR spectra in Fig. 2f, peaks assigned to C−O stretching (wavenumber of 1205, 1145, and 1068 cm$^{-1}$) disappeared after the calcination of the deposited MeO-AlO$_x$ film at 600 °C. Hence,

the calcination temperature was determined to be 600 °C for membrane fabrication.

The −OCH$_3$ group density is mainly controlled by the CH$_3$OH and H$_2$O dosage. It is noteworthy that even when CH$_3$OH was overdosed (pulse pressure of 0.40 Torr compared to 0.06 Torr of H$_2$O), the atomic ratio of carbon within the deposited film was only 3.5% (characterized by XPS; Fig. 2g). Considering that each carbon atom is surrounded by ~19.8 oxygen atoms and ~9.5 aluminum atoms within this film (Supplementary Fig. 8), continuous transport pathways can hardly be formed even if all the −OCH$_3$ groups are burned off. This low density of −OCH$_3$ groups can be attributed to either the greater reactivity of H$_2$O compared to the CH$_3$OH molecules, or the replacement of −OCH$_3$ groups by −OH groups according to Eq. 3. Therefore, to further increase the −OCH$_3$ group density, the H$_2$O dosage was adjusted. As summarized in Fig. 2g, when the H$_2$O pulse pressure was decreased from 0.06 (i.e., the typical H$_2$O pulse for the Al$_2$O$_3$ ALD deposition) to 0.007 Torr, the atomic ratio of carbon within the deposited films

increased from 3.5% to 10.0%. Specifically, for the MeO-AlO$_x$ film deposited with 0.007 Torr of H$_2$O dosage, each carbon atom is surrounded by ~6.1 oxygen atoms and ~ 2.9 aluminum atoms (Supplementary Fig. 8), which is more promising for the creation of continuous transport pathways within the membrane compared to the film deposited with 0.06 Torr of H$_2$O dosage.

## Transport and separation properties of the aluminum oxide membranes

The transport properties of the fabricated AlO$_x$ membranes were evaluated using a customized two-chamber diffusion cell (Fig. 3a). Specifically, water permeation was characterized to rationalize our design and verify the formation of continuous voids within the calcinated AlO$_x$ films. As shown in Fig. 3b, a notably higher water permeance was observed for the calcinated AlO$_x$ membranes (2.6 × 10$^{-3}$ LMH bar$^{-1}$) compared to the ALD deposited MeO-AlO$_x$ films (both deposited with 0.007 Torr of H$_2$O dosage), suggesting that transport

pathways can be created via burning off the decorated −OCH$_3$ groups. No apparent water flux was observed for the calcinated AlO$_x$ membranes created with 0.06 Torr of H$_2$O dosage. Therefore, the pulse pressure during membrane construction was set at 0.4 and 0.007 Torr for CH$_3$OH and H$_2$O dosage, respectively. Densification caused by the phase transition of thin films at high temperatures, i.e., from amorphous to crystalline, has been reported to contribute to the pore formation as well[38]. However, this is not the main mechanism here as no water transport was observed with the Al$_2$O$_3$ ALD films calcinated at the same temperature (600 °C; Fig. 3b).

The selectivity of the fabricated AlO$_x$ membranes for the transport of monovalent and divalent salts was then evaluated using the same concentration-gradient-driven process. Single salt solutions (NaCl, CaCl$_2$, and Na$_2$SO$_4$) were placed on the concentrated side and the salt permeation rate was calculated by monitoring the increase in electric conductivity on the dilute side. As summarized in Fig. 3c, a significantly higher permeation rate was observed for NaCl over the permeation

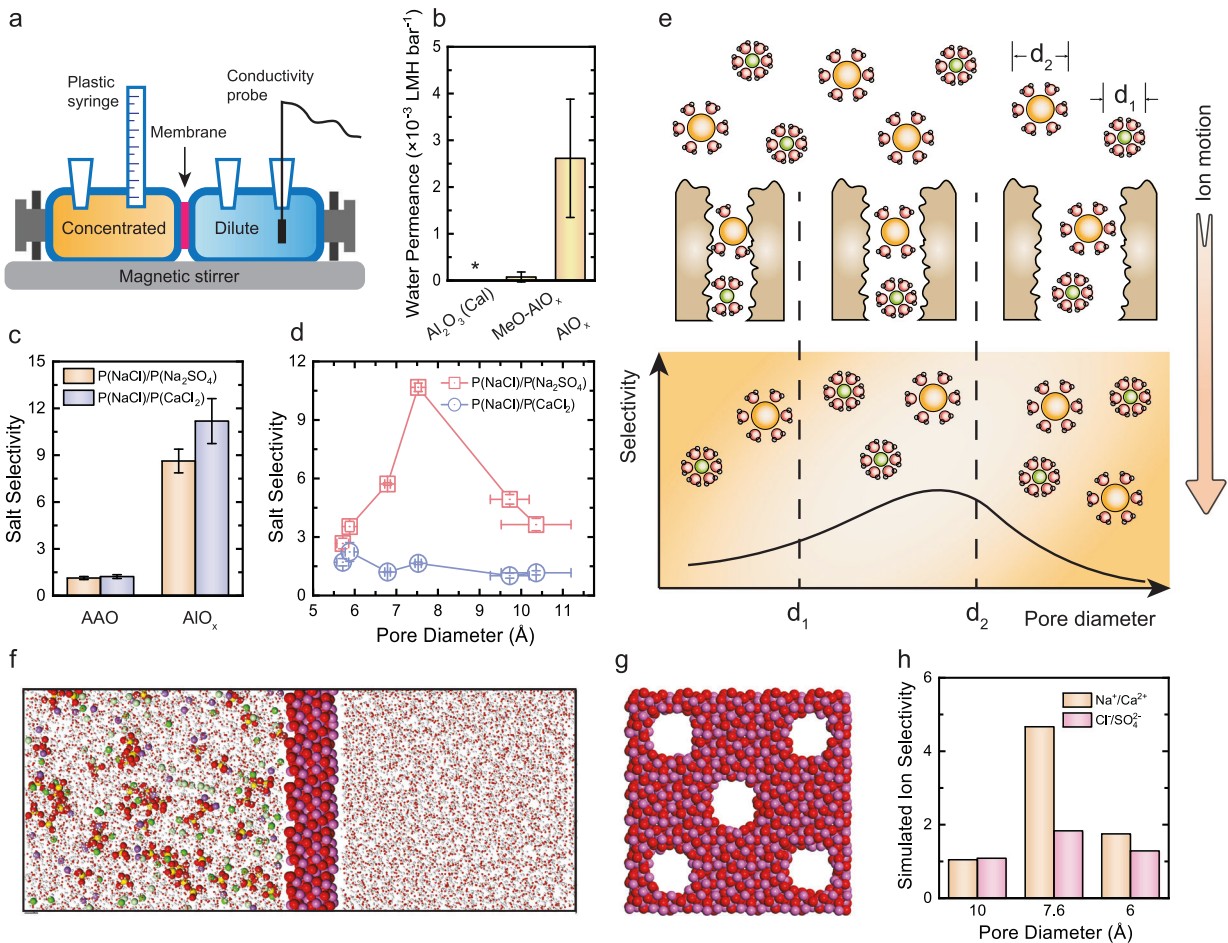

**Fig. 3 | Selective ion transport through the AlO$_x$ ALD membranes with sub-nanometer pores. a** Schematic diagram illustrating the customized diffusion cell used for the transport tests. **b** Water permeability coefficient of the calcinated AlO$_x$ membranes (right), the deposited MeO-AlO$_x$ films (middle), and the calcinated Al$_2$O$_3$ membranes deposited by the conventional Al$_2$O$_3$ ALD deposition (left). Error bars represent standard deviations from measurements with three different membranes. **c** Selectivity between NaCl and Na$_2$SO$_4$, and NaCl and CaCl$_2$, determined from salt permeation through the AAO substrates and the composite AlO$_x$ membranes. 0.1 M single salt and DI water were used as the concentrated and dilute solutions, respectively. Error bars represent standard deviations from measurements with three membranes. **d** Dependence of the selectivity between monovalent salts (i.e., NaCl) and divalent salts (i.e., Na$_2$SO$_4$ and CaCl$_2$) on the size of subnanometer pores. Polyamide TFC membranes with varied pore diameters were

mounted into the diffusion cell. For these experiments, 0.1 M single salt solutions were used as the concentrated solutions, whereas 0.2 and 0.3 M sucrose solutions were used as the dilute solutions to balance the osmotic pressure during the measurements with monovalent and divalent salts, respectively. Error bars represent standard deviations from measurements with three membranes. **e** Schematic diagram illustrating the selective transport of hydrated ions through pores of varied sizes. **f** Schematic of the simulation box consisting of a salt solution with 1.0 M of Na$^+$, Ca$^{2+}$, Cl$^-$, and SO$_4^{2-}$ ions (left), an AlO$_x$ ALD membrane (middle), and pure H$_2$O (right). **g** Cross-sectional view of an AlO$_x$ ALD membrane with 7.6 Å pores. Five pores were drilled such that significant ion flux can be obtained in a concentration-gradient-driven process. **h** Simulated monovalent and divalent selectivity (Na$^+$/Ca$^{2+}$ and Cl$^-$/SO$_4^{2-}$) through AlO$_x$ membranes with pore diameters of 6, 7.6, and 10 Å.

rate of $Na_2SO_4$ and $CaCl_2$ (i.e., salt selectivities of 8.6 and 11.2, respectively). As a comparison, the pristine AAO substrates showed limited capability in differentiating the transport of these solutes, and a ratio of ~1.2:1 was observed between the NaCl flux and the flux of the divalent salts (Fig. 3c). The slightly faster NaCl permeation through the nanopores of the AAO substrates is attributable to the higher diffusion coefficients of monovalent ions ($1.334 \times 10^{-9}$ and $2.032 \times 10^{-9}$ $m^2 s^{-1}$ for $Na^+$ and $Cl^-$ ions, respectively[39]) compared to divalent ions ($7.9 \times 10^{-10}$ and $1.065 \times 10^{-9}$ for $Ca^{2+}$ and $SO_4^{2-}$ ions, respectively[39]). Thus, the observed fast NaCl permeation with the constructed $AlO_x$ TFC membranes is contributed by the $AlO_x$ active layer.

Electrostatic interactions have been proposed to govern the selective transport of ions with different charges through nanopores[40]. For example, divalent cations (e.g., $Ca^{2+}$) were observed to permeate faster through negatively charged pores compared to the monovalent cations (e.g., $Na^+$), which was explained by the higher electrostatic attraction between the divalent cations and the pores[40]. The permeation of divalent cations became slower when pores were positively charged[41]. This electrostatic effect can explain our observed faster $Na_2SO_4$ permeation through the $AlO_x$ membranes compared to $CaCl_2$ (i.e., lower NaCl/$Na_2SO_4$ selectivity compared to NaCl/$CaCl_2$ selectivity in Fig. 3c), as $AlO_x$ is positively charged at the experimental pH (5.7)[42,43]. Nevertheless, an 8.6-time-higher NaCl flux was observed compared to $Na_2SO_4$ flux, suggesting that electrostatic interaction is not the dominant mechanism contributing to the observed selective ion transport. In other words, the ultrahigh monovalent/divalent selectivity obtained with the $AlO_x$ membranes should be attributed to the steric confinement within the active layer.

To further demonstrate the role of pore size in differentiating the transport of monovalent and divalent salts, the permeation rate of NaCl, $CaCl_2$, and $Na_2SO_4$ through polyamide (PA) TFC membranes with different pore diameters was measured. It is worth noting that although the observed selectivity with the fabricated $AlO_x$ membranes cannot be fully interpreted by the experiments with PA TFC membranes, the PA TFC membranes, ranging from reverse osmosis (RO) to ultrafiltration (UF), provide a unique platform to study the size-dependent transport phenomenon at the subnanometer scale. Notably, the highest selectivity between $Cl^-$ and $SO_4^{2-}$ ions was obtained with the 7.5 Å pores (i.e., NaCl/$Na_2SO_4$ selectivity of 10.7, Fig. 3d); this selectivity declined for membranes with smaller pores and membranes with larger pores. This observed dependence of ion-ion selectivity on the pore size is different from our expectation, where a monotonic increase of the selectivity should be obtained with enhanced confinement (i.e., with smaller pore size) due to the higher extent of ion dehydration. Considering that $SO_4^{2-}$ and $Cl^-$ ions have respectively hydrated diameters of 7.6 and 6.6 Å[44,45], this observed selectivity trend suggests that the state of hydration/dehydration, rather than the extent of dehydration (i.e., the number of surrounded $H_2O$ molecules lost), plays the predominant role in determining the selective ion transport (Fig. 3e)[46].

Specifically, in the case of ion permeation through pores larger than $SO_4^{2-}$ ions, both $SO_4^{2-}$ and $Cl^-$ ions permeate quickly with an intact hydration shell, resulting in a limited ion-ion selectivity[47]. For ions permeating through pores similar to the size of the $SO_4^{2-}$ ions, $SO_4^{2-}$ ions have to adjust their hydration shell and thus need to overcome a high resistance during both pore entry and diffusion inside the pore. However, the smaller $Cl^-$ ions can still permeate through the pore rapidly without dehydration, leading to a high $Cl^-$/$SO_4^{2-}$ selectivity[47]. When the pores are smaller than the size of $Cl^-$ ions, both $SO_4^{2-}$ and $Cl^-$ ions must undergo significant dehydration and the selectivity between their transport is limited[47]. This pore-size-dependent dehydration-based selective ion transport only holds when there is no specific interaction between the pore wall and ions. Notably, despite $Ca^{2+}$ ions (diameter of 8.2 Å[48]) being more prone to dehydration compared to $Na^+$ ions (diameter of 7.0 Å[48]), limited selectivity between NaCl and

$CaCl_2$ was observed with the PA TFC membranes as the strong interaction of $Ca^{2+}$ with $-COO^-$ functional groups in the PA membrane can compensate the dehydration energy of $Ca^{2+}$ ions. This size-dependent ion selectivity also suggests that the average pore diameter of the fabricated $AlO_x$ membranes was ~7.6 Å, given that these membranes showed an exceptional capability in differentiating the transport of $SO_4^{2-}$ and $Cl^-$ ions and the transport of $Ca^{2+}$ and $Na^+$ ions.

Molecular dynamics (MD) simulations were then performed to verify the dominant role of ion hydration/dehydration in differentiating the transport of monovalent and divalent ions. Salt solutions containing $Na^+$, $Ca^{2+}$, $Cl^-$, and $SO_4^{2-}$ ions and pure $H_2O$ were placed on two sides of the $AlO_x$ membrane (Fig. 3f). Five pores with identical diameters were drilled to facilitate the ion flux in a concentration-gradient-driven process (Fig. 3g). The selectivity between monovalent and divalent ions was calculated for $AlO_x$ pores with diameters of 10, 7.6, and 6 Å to study the size-dependent ion selectivity.

When the pore diameter was 10 Å, all four ions ($Na^+$, $Ca^{2+}$, $Cl^-$, and $SO_4^{2-}$) retained a similar hydration shell within the pores compared to their respective hydration shell in the bulk solution (Supplementary Fig. 10). Consequently, the selectivity between monovalent and divalent ions was close to 1 ($Na^+/Ca^{2+} = 1.0$ and $Cl^-/SO_4^{2-} = 1.1$; Fig. 3h). When the pore diameter was narrowed to 7.6 Å, divalent ions partially lost surrounding $H_2O$ molecules during their passage through the membrane pores while monovalent ions still maintained their original hydration shell (Supplementary Fig. 10), leading to an increased monovalent/divalent selectivity ($Na^+/Ca^{2+} = 4.7$ and $Cl^-/SO_4^{2-} = 1.9$; Fig. 3h). Further narrowing the pore diameter to 6 Å enforced both the monovalent and divalent ions to adjust their hydration shell (Supplementary Fig. 10), resulting in a lower selectivity between them ($Na^+/Ca^{2+} = 1.8$ and $Cl^-/SO_4^{2-} = 1.3$; Fig. 3h).

## Tuning the pore size of the aluminum oxide membranes for molecular sieving

Pore structure within the constructed $AlO_x$ membranes was adjusted by varying the co-reactants during the monofunctional-alcohol-modulated ALD deposition. Specifically, ethanol ($CH_3CH_2OH$; 0.2 Torr) and isopropanol (($CH_3)_2CHOH$; 0.2 Torr) were employed to incorporate ethoxy ($-OCH_2CH_3$) and isopropoxy ($-OCH(CH_3)_2$) groups into the deposited films, respectively (Fig. 4a, b). These materials were deposited on top of the AAO substrates to construct pinhole-free films. We then calcinated the hybrid films to burn off carbonaceous species of different dimensions to create active layers with varied pore sizes. According to Fig. 4c, when the co-reactants were switched from $CH_3OH$ to $CH_3CH_2OH$ and $(CH_3)_2CHOH$, the water permeability of the fabricated membranes increased from $2.6 \times 10^{-3}$ to $3.5 \times 10^{-2}$ and $5.1 \times 10^{-2}$ LMH $bar^{-1}$, respectively. The enhanced water permeability suggests that larger pores were created when we burned off the $-OCH_2CH_3$ and $-OCH(CH_3)_2$ groups within the deposited films.

To further estimate the membrane pore size variation when using different alcohols as co-reactants during the deposition and demonstrate the tunable capabilities of the $AlO_x$ membranes in separating solute transport, the permeation rate of neutral solutes of various sizes through each fabricated membrane was measured. Neutral solutes were used here to exclude the impact of other mechanisms (e.g., Donnan and dielectric[49]) on the solute transport such that the impact of size can be explicitly demonstrated. A hyperbolic dependence of solute flux on their radius was observed with the membranes created with $CH_3OH$, $CH_3CH_2OH$, and $(CH_3)_2CHOH$, consistent with the size-exclusion-dominating selective solute transport (Fig. 4d).

For the $AlO_x$ membranes created with $CH_3OH$ (Fig. 4d, e), limited transport was observed when the solute diameter is larger than 8.1 Å (sucrose of 8.2 Å and raffinose of 10.0 Å). When the solute diameter is smaller than 6.6 Å (glucose of 6.5 Å, erythritol of 5.3 Å, and ethylene glycol of 4.0 Å), these solutes can transport through the membrane pores, and their permeation rate increased with the decreasing solute

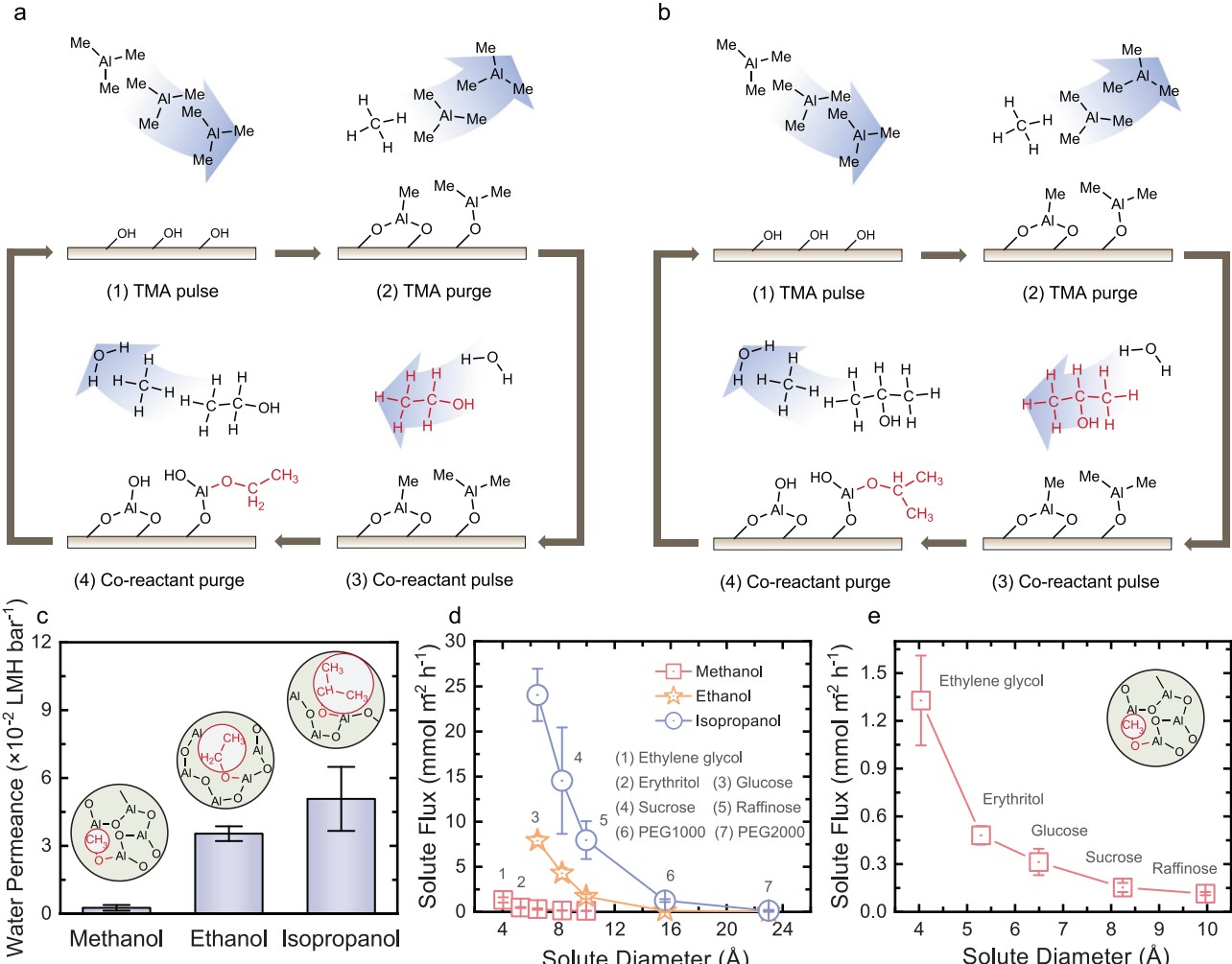

**Fig. 4 | Tuning the subnanometer pores of the AlO$_x$ membranes for molecular sieving.** Schematic diagrams of **a** ethanol-modulated and **b** isopropanol-modulated ALD deposition. Both processes consist of four steps: (1) TMA pulse; (2) TMA purge; (3) co-reactant pulse (a mixture of H$_2$O and ethanol for ethanol-modulated process, and a mixture of H$_2$O and isopropanol for isopropanol-modulated process); and (4) co-reactant purge. **c** Water permeability coefficient of the AlO$_x$ membranes fabricated with methanol, ethanol, and isopropanol as co-reactants. In each measurement, 1 M NaCl was utilized as the concentrated solution and DI water was used on the dilute side. Error bars represent standard deviations from measurements with three membranes. **d** Permeation rate of neutral solutes through the AlO$_x$ membranes with tunable subnanometer pores. For each test, 0.1 M single solute solutions were placed on the concentrated side and the permeation rate was calculated by measuring the TOC increase on the permeate side. Error bars represent standard deviations from measurements with at least two membranes. **e** Amplification of the solute flux through the AlO$_x$ membranes fabricated with methanol in **d**.

diameter. The observed solute flux suggests that the average pore diameter of the membranes fabricated with CH$_3$OH is between 6.6 and 8.1 Å, consistent with the average pore diameter estimated based on the selective transport between monovalent and divalent ions (~7.6 Å).

The flux of neutral solutes through the AlO$_x$ membranes increased when CH$_3$CH$_2$OH was utilized as the co-reactant compared to CH$_3$OH. For instance, the flux of glucose increased from 0.31 to 7.9 mmol m$^{-2}$ h$^{-1}$. More significantly, sucrose and raffinose can permeate through the AlO$_x$ membranes fabricated with CH$_3$CH$_2$OH with respective fluxes of 4.3 and 1.7 mmol m$^{-2}$ h$^{-1}$, confirming that larger pores were created by burning out the −OCH$_2$CH$_3$ groups. We note that negligible permeation of polyethylene glycol 1000 (PEG 1000) was observed with the membranes created with CH$_3$CH$_2$OH, indicating that the average pore diameter of these membranes is between 10.1 and 15.5 Å.

When (CH$_3$)$_2$CHOH was used as the co-reactant, the solute flux through the constructed AlO$_x$ membranes further increased. For example, the flux of raffinose increased from 1.7 to 8.0 mmol m$^{-2}$ h$^{-1}$ when the co-reactants were switched from CH$_3$CH$_2$OH to (CH$_3$)$_2$CHOH. Furthermore, PEG 1000 was observed to permeate through the membranes fabricated with (CH$_3$)$_2$CHOH with a flux of

1.3 mmol m$^{-2}$ h$^{-1}$. This neutral solute transport measurement suggests that the average pore size of the AlO$_x$ membranes constructed with (CH$_3$)$_2$CHOH is between 15.7 and 23.0 Å. Overall, the observed dependence of solute flux on their size shows the feasibility of using this proposed ALD strategy to construct membranes with tunable pore sizes for molecular sieving.

## Discussion

We demonstrated the construction of AlO$_x$ membranes with tunable subnanometer pores by burning off the carbonaceous functional groups within the hybrid films prepared using a monofunctional-alcohol-modulated ALD process. This molecular-level design of the pore structure enables the application of ceramic membranes in precise ion separations with subnanometer confinement. Specifically, the AlO$_x$ membranes fabricated with CH$_3$OH can selectively transport NaCl 8.6 times faster compared to Na$_2$SO$_4$, matching the highest selectivity obtained with state-of-the-art polyamide membranes (10.7:1 with NF270). The observed slower passage of Na$_2$SO$_4$ is attributable to the enforced dehydration of SO$_4^{2-}$ ions within the membrane pores. As a comparison, Cl$^-$ ions can permeate through the pores quickly with an

intact hydration shell. Further adjusting the co-reactants during the ALD deposition creates $AlO_x$ membranes with varied pore sizes, enhancing the versatility of ceramic membranes for precise molecular sieving. Future work could focus on using other metal precursors to construct ceramic membranes with varied surface chemistry. Such membranes will enable the systematic investigation of the role of ion-membrane chemical interactions in selective ion transport.

## Methods

### Materials and chemicals

AAO templates (pore diameter between 20 and 30 nm according to the manufacturer, 60 μm in thickness) were purchased from Hefei Pu-Yuan Nano Technology Co., Ltd. (China). Commercial SW30 XLE, XLE, NF90, NF270, UA60, and NDX membranes were purchased from Sterlitech Corporation (Auburn, WA). TMA (99.999%), $CH_3OH$ ($\geq 99.9\%$), $CH_3CH_2OH$ ($\geq 99.9\%$), $(CH_3)_2CHOH$ ($\geq 99.9\%$), NaCl ($\geq 99.5\%$), $Na_2SO_4$ ($\geq 99.0\%$), $CaCl_2$ ($\geq 99.0\%$), EG ($\geq 99.0\%$), erythritol ($\geq 99.0\%$), glucose ($\geq 99.0\%$), sucrose ($\geq 99.5\%$), raffinose ($\geq 98.0\%$), PEG 1000, and PEG 2000 were purchased from Sigma-Aldrich (St. Louis, MO). Deionized water ($>18.2$ MΩ cm) was obtained from a Milli-Q ultrapure water purification system (Integral 10, Millipore, Billerica, MA).

### Aluminum oxide thin-film composite membrane fabrication

AAO substrates were loaded as received into the ALD reaction chamber (Fiji G2, Veeco-CNT) and preheated at 150 °C for 30 min. The ALD deposition of carbon-decorated aluminum oxide materials was conducted with the sequential pulse of TMA and the co-reactant consisting of a mixture of monofunctional alcohol ($CH_3OH$, $CH_3CH_2OH$, or $(CH_3)_2CHOH$) and water. Argon (99.999997%) gas was supplied to carry the precursors into the reaction chamber and to purge the chamber to remove the unreacted precursors after each step of the reaction. The TMA pulse pressure was fixed at 0.25 Torr, and the dosage of alcohol and water was adjusted. All the precursors were maintained at room temperature during deposition. This cycle was repeated to construct a continuous film (17 nm in thickness) on top of the AAO substrates. Porous $AlO_x$ layers were then generated by calcinating the deposited hybrid layers in a muffle furnace (Thermolyne, Thermo Fisher Scientific, Waltham, MA) for 5 h under an air atmosphere. The ramp rate was maintained at 1 °C min$^{-1}$ and the calcinating temperature was optimized between 400 and 650 °C to burn off the carbonaceous species within the deposited film.

### Material characterization

In situ FTIR measurement was performed in transmission mode on a Nicolet 6700 FTIR spectrometer (Thermo Scientific) interfaced with a custom-built viscous flow ALD reactor. $ZrO_2$ powders ($<100$ nm diameter) pressed into a stainless-steel mesh (Fotofab, Chicago, IL) were used as the substrates for FTIR analysis. The mesh was mounted into the FTIR stage and equilibrated inside the ALD reactor before the FTIR measurement. The growth rate of the ALD material and the morphology of the constructed $AlO_x$ composite membrane were characterized via scanning electron microscopy (SEM; SU8230, Hitachi, Japan). For growth rate characterization, SEM images of the AAO substrates undergoing different cycles of ALD modification were captured (500k magnification) and the pore radius of each sample was analyzed using ImageJ (averaged from >20 measurements). The film growth rate was calculated by monitoring the dependence of the AAO pore radius on the number of ALD cycles conducted. The elemental composition of the ALD deposited film and the film undergoing calcination was characterized by XPS spectroscopy (VersaProbe II, Physical Electronics, Chanhassen, MN) using Al Kα radiation (hv = 1486.6 eV). To eliminate the impact of the adventitious carbon on the analysis, Ar sputtering was conducted with a power of 1 kV before the XPS measurement. The binding energy of each element was calibrated by setting the binding energy of the O 1s peak to 530.0 eV.

### Membrane transport test

The transport properties of the fabricated $AlO_x$ membranes were evaluated using a homemade diffusion cell (Fig. 3a). Each chamber of the cell has a volume of ~60 mL. A membrane coupon with an effective area of 0.64 cm$^2$ was assembled between these two chambers with the $AlO_x$ active layer facing the concentrated side. To test the water permeation of the fabricated composite membranes, 1 M NaCl and DI water were used as the concentrated and dilute solutions, respectively. Water flux was measured by monitoring the volume increase of the concentrated chamber using a graduated plastic syringe. The ability of the fabricated $AlO_x$ membranes to differentiate the transport of different solutes was evaluated by using 0.1 M solutes as the concentrated solutions (i.e., NaCl, $CaCl_2$, $Na_2SO_4$, erythritol, xylose, glucose, sucrose, raffinose, PEG 1000, and PEG 2000). Salt concentration in the permeate side was monitored using the conductivity probe, and the concentration of neutral solutes was measured by a total organic carbon analyzer (TOC-VCSH, Shimadzu, Japan). All measurements were conducted at 22 °C with vigorous mixing in both chambers.

### Molecular dynamics simulations

The simulation box consisted of three main components: an amorphous $Al_2O_3$ membrane with subnanometer pores, a salt solution containing $H_2O$, $Na^+$, $Ca^{2+}$, $Cl^-$, and $SO_4^{2-}$ ions, and pure $H_2O$. To create the amorphous $Al_2O_3$, a sample of crystalline $Al_2O_3$ (downloaded from the Materials Project) was heated at 3000 K for 100 ps using Materials Studio in Forcite Module with COMPASSII as the force field. The amorphous $Al_2O_3$ was then generated using the *supercell* function. The salt solution and pure $H_2O$ were constructed using Amorphous Cell and combined with the amorphous $Al_2O_3$ using the *build* function.

The molecular structure of the simulation box was optimized using the Forcite Module with COMPASSII as the force field. During the geometric optimization, the convergence threshold for maximum energy change, maximum force, and maximum displacement were set to 0.001 kcal mol$^{-1}$, 0.5 kcal mol$^{-1}$ Å$^{-1}$, and 0.015 Å, respectively. To release the internal stress in the system, an MD simulation was performed under the NPT ensemble for 100 ps at 0.0001 GPa and 298 K until the density was stable over time. Further optimization of the system involved an MD simulation under the NVT ensemble for 100 ps. The simulation of ion passage involved the calculation of electrostatic interaction using Ewald and van der Waals force using Atom base. Precise Nose-Hoover temperature control mode and Berendsen pressure control mode were used in the simulation, and the concentration of ions permeating the membranes was calculated at the end of the MD simulation. All the simulations were conducted at the temperature of 298 K with a step size of 0.5 fs.

## Data availability

The data that support the findings of this study are available from the corresponding authors upon request.

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

## Acknowledgements
This work was primarily supported as part of the Center for Enhanced Nanofluidic Transport (CENT), an energy frontier research center funded by the U.S. Department of Energy, Office of Science, Basic Energy Sciences, under Award No. DE-SC0019112 (M.E. and X.Z.). The authors also gratefully acknowledged the financial support from the Israel-U.S. Collaborative Water-Energy Research Center (CoWERC), supported by the Binational Industrial Research and Development Foundation under Energy Center grant EC-15 (R.S., A.U.M., J.W.E., and J.-H.K.).

## Author contributions
X.Z., J.-H.K., and M.E. conceived the idea. X.Z., R.S., D.H., T.C., and X.S. designed and performed the experiments and analyzed the data. All authors discussed the results and wrote the manuscript.

## Competing interests
The authors declare no competing interests.
