## [Peer Review File · Nature Communications]

Reviewers' Comments:

Reviewer #1:

Remarks to the Author:

The idea, ALD of organic-containing oxides followed by calcination to convert the organic moieties into micropores, thus producing composite membranes with the microporous layers as the selective layers, has been reported by a number of groups a few years ago. The only difference of the present work is that this group of authors included an additional pulse of alcohols into the ALD process. However, the authors failed to show the advantage of their modification to previously reported method. By no means I can support the publication of such a work in a decent journal like nature comm:

1. The novelty of the authors method and the corresponding advantages were not clear;
2. The separation performances of their membranes were much poorer than that of conventional polyamide membranes in terms of both rejection and permeability;
3. Some important conclusions are lack of solid experimental evidence. For example, the authors mentioned "dehydration of the large divalent ions within the subnanometer pores", simulation or experimental evidences are required;
4. The membranes were not sufficiently characterized. For instance, the pore size and pore size distribution, porosities, and the surface hydrophilicity were unknown although they played significant roles in determining the separation performances. The calcined membranes took a grey color, implying there should be a considerable amount of carbon in the resulted porous layers; and the layers are expected to show somewhat a hydrophobic surface, which will hinder water permeance;
5. The authors spent many words to discuss the advantage of ceramics. However, their membranes synthesized in this work are not ceramics; they were just amorphous aluminum oxides with some organic impurities;
6. The separation tests were performed in a concentration-diffusion mode. From the standpoint of practical applications, one would like to see the separation performances operated under pressures;
7. The authors are over exaggerated when commenting their results. For example, "for the first time the application of ceramic membranes in precise ion separations" in the Discussion part. Do the authors really have no aware of previous report on the application of ceramic membranes in ion separations?

Reviewer #2:

Remarks to the Author:

In this work, authors developed an ALD-based process to fabricate ceramic membranes with tunable subnanometer pores and demonstrated its use for precise ion separation.; excellent separation of NaCl from Na₂SO₄ has been shown, resulting from enforced dehydration of SO₄²⁻ and thus slower transport of Na₂SO₄. Adjustment of ALD parameters can be used to further tune the membrane pores for precise molecular separation. This is a novel work and is expected to have significant impact on ceramic membranes for precise separation. I would recommend publication after major revision for addressing the following questions and comments:

- 1) Ceramic membranes are more expensive compared to traditional polymeric membranes. However, one major advantage of ceramic membranes is that it can be used for applications where polymeric membranes would fail. How is using ceramic membranes justified for an application involving water where polymeric membranes can be readily used?
- 2) How do you prove that there is uniformity in distribution of molecular scale sacrificial fragments? The authors claim that the uniformity is necessary to eliminate pore collapse. And they ensure that there is uniformity of molecular scale sacrificial elements by using a low vapor pressure precursor (i.e., methanol). How do they prove this uniformity? Maybe a pore size distribution measurement after calcination would show a sharp, possibly more uniform sized pores. This can be used against a case where there is condensation (possibly carrying out the modified ALD reaction at low temperature to allow condensation), and the pore size distribution is wider.

3) In MLD and ALD processes, usually the organic precursor is so chosen that after it reacts with the metal precursor, it ends up having one reactive end exposed (e.g., hydroxyl, amine etc.) This is done to ensure enough reactive sites for the next round of metal precursor. In this work, methanol was used along with water. The use of methanol limits the possibility of exposure to reactive groups in the next round of metallic precursor deposition. However, there is water that ensures reactive groups for the next cycle. Is the Methanol purged first followed by Water? Does the water react with unreacted Al-CH₃ groups, or it replaces Al-O-CH₃ groups? Why is some unreacted Al-CH₃ group remains after methanol purge? And if Water replaces Al-O-CH₃ groups, why doesn't it replace all of them? Also, mixing up two reactant (one with terminating group and another with reacting group) increases the chance of non-uniformity where there might be regions of carbon containing groups in the hybrid material and regions containing no hybrid groups. This may cause formation of pores that are not connected or are non-continuous, connecting the feed and permeating side.

4) It is important to mention in the main text, the number of cycles needed to form a thin film on the AAO substrate.

5) How was it ensured that the hybrid thin film formed is defect-free before organic removal and does not have any defects that are selective and provide ion selectivity?

6) It is shown that changing the organic precursor has an effect on the tunability of the membrane; larger organic precursors make larger pores with higher permeance. Does the change of H₂O and methanol pulse pressure impart any tunability?

7) The authors claim that a certain pulse pressure of methanol and water was promising to create continuous transport pathways. Why is this? What was the rationale behind choosing "0.4 and 0.007 Torr for CH₃OH and H₂O dosage, respectively"?

8) What was the temperature for calcination of the hybrid nanofilms? Was there any residual carbon in the film after calcination?

9) The calcined hybrid film's surface charge should be measured. Because the AlO_x film in question might have carbon residue and show some alteration of surface charge than that of AlO_x films reported in literature.

10) Why was separation carried out at pH 5.7? is the pH for the real salt containing wastewater? Or it was randomly chosen?

11) The salt rejection experiments using the commercial membranes is to establish the role of pore size in providing monovalent/divalent selectivity. While this is important to explain the monovalent/divalent selectivity, the experiment with commercial membrane is too distracting for readers in the main text. This can be moved to the SI.

12) For the ions, diameter was used to explain the possible pore size of the membrane. It is suggested that the authors also use diameter for neutral solutes (lines 323-330).

13) What was pulse pressure when other co-reactants (C₂ and C₃ alcohols) were used? Detailed methodology is essential. Also, it is mentioned that the calcination temperature was optimized; how was this optimized? What was the goal of the optimization? Least time? Or a certain pore size? Or something else. These details are missing.

14) Is it possible to prove experimentally or computationally the "possible dehydration of one ion to obtain selectivity" theory? It seems necessary to have some evidence for the ion dehydration claim. If it is indeed dehydration requirement for one ion to pass, which makes it difficult for it to pass through the pores, does that mean that the selectivity between the two ions observed under diffusion system would not be seen if the permeation was done under pressure (say at pressures higher to overcome this dehydration energy requirement? How much pressure would it be needed to eliminate the difficulty of dehydration?

Reviewer #3:

Remarks to the Author:

This paper reports the preparation of subnanometer porous alumina selective layer supported on AAO substrate by using the monofunctional-alcohol-modulated atomic layer deposition method. The subnanometer pores are created by burning off organic moieties. This method is elegant and novel and provides an effective way to accurately control the pore size of ceramic membrane. The membranes are well characterized and show high monovalent to divalent ion selectivity. The manuscript is well written, and can be accepted for publication after some minor revision.

After the ALD process, the membranes are calcined in air to generate pores. It would be better to perform control experiments that the membranes are calcined in inert gas such as argon at the same temperatures to see how the membrane structure changes. This will help understand the pore formation mechanisms.

Response to Reviewers' Comments

In the following response letter, black italic type represents the exact comments from the reviewers, blue type represents our response, green type represents unmodified text in the manuscript, and bold green type represents new text added to the revised manuscript. Lines listed refer to the revised manuscript.

Reviewer #1

General Comment The idea, ALD of organic-containing oxides followed by calcination to convert the organic moieties into micropores, thus producing composite membranes with the microporous layers as the selective layers, has been reported by a number of groups a few years ago. The only difference of the present work is that this group of authors included an additional pulse of alcohols into the ALD process. However, the authors failed to show the advantage of their modification to previously reported method. By no means I can support the publication of such a work in a decent journal like nature comm:

Response We thank the reviewer for his/her/their time and effort in reviewing our manuscript. Reviewer 1 mainly questioned the novelty of our work. We hope to address the reviewer's concern in our detailed responses presented below.

Comment The novelty of the authors method and the corresponding advantages were not clear:

I
Response We thank the reviewer for this opportunity to clarify the novelty of our fabrication strategy and the corresponding advantages. The reviewer is correct that the idea of using molecular layer deposition (MLD) to construct organic-containing metal oxides followed by calcination to produce membranes with microporous selective layers has been reported by other groups before. We acknowledged these works in the Introduction of our manuscript:

Lines 75-80: "One proposed strategy is calcinating the metal-organic hybrid films prepared using the molecular layer deposition (MLD) technique^{1,2,3}. Specifically, in these MLD processes, metal atoms and organic linkers are deposited on the substrate surface, layer by layer, by sequentially exposing the substrates to the metal-organic precursors (e.g., titanium tetrachloride) and the bifunctional co-reactants (mainly ethylene glycol, EG). By burning off the organic linkers within the hybrid films, (sub)nanometer pores were supposed to be generated."

We also pointed out the limitations of the previous strategy:

Lines 80-83: "Despite the attempted molecular-level design, the fabricated ceramic membranes showed limited capability in differentiating between the transport of different ions³. This inconsistency is attributable to the condensation of EG during the MLD deposition, creating large defects after calcination⁴."

Specifically, studies found that pores with a diameter of ~3.8 nm were created after calcinating the hybrid films prepared by an EG-involved MLD process⁴. The formation of large pores was attributed to the condensation of EG molecules on the substrate surface during the MLD process, leading to multimolecular layers⁴. The existence of large pores (defects) within the selective layers deteriorated the separation capability of the constructed membranes. For instance, one study examined the selectivity of EG-based MLD membranes towards the permeation of Na₂SO₄ and

NaCl, and a selectivity of only 1.5:1 was achieved³. This limited selectivity between monovalent and divalent ions with the presence of large pores is consistent with our observed size-dependent ion selectivity (Figure 3D).

Therefore, to improve the selectivity of these ceramic membranes and employ them for precise separation, we need to avoid the creation of large defects. Unfortunately, this is not feasible with the previous strategy of combining the conventional MLD deposition and post-deposition calcination as even EG, one of the most volatile multifunctional organic molecules, condenses during the MLD deposition. To address this limitation, we proposed a brand-new monofunctional-alcohol-modulated atomic layer deposition (ALD) process to incorporate distributed molecular-scale organic decorations into the metal oxide films. **It is worth highlighting that our proposed ALD process is conceptually different from the conventional ALD deposition in which all the precursors should be multifunctional.** Avoiding using the low-volatility multifunctional organic precursors theoretically should lead to better control of the pore structure within the fabricated selective layers. To emphasize the novelty of our deposition process, we added the following sentence to our manuscript:

Lines 83-87: “Hence, to realize the application of this sacrificial template strategy in constructing **defect-free** ceramic membranes with subnanometer pores, it is imperative to build hybrid films with distributed molecular-scale sacrificial segments. **However, this objective can barely be realized with the conventional MLD process as EG is among the most volatile bifunctional co-reactants.**”

The transport test verifies our creation of ceramic membranes with enhanced separation capability. Specifically, the proposed fabrication strategy enabled us to construct selective layers with a diameter of ~0.76 nm, which can barely be achieved with the previous MLD process. This precise pore size control led to a selectivity of 8.6:1 between the transport of Na₂SO₄ and NaCl, outperforming the selectivity obtained with the previous MLD membranes and matching the highest selectivity achieved with the polyamide membranes. Furthermore, with the capability of accurately controlling the pore structure within the selective layers, we also demonstrated the tuning of pore size at the angstrom level (Figure 4), which is crucial to the applications of precise separation.

Comment 2 *The separation performances of their membranes were much poorer than that of conventional polyamide membranes in terms of both rejection and permeability;*

Response We thank the reviewer for the comment. The reviewer is correct that polyamide membranes are the gold standard for desalination applications due to their outstanding salt rejection capability and water permeability. However, solute-solute separation, rather than water-solute separation, was mainly discussed in this manuscript. Specifically, we demonstrated that with the proposed fabrication strategy, we were able to construct AlO_x membranes showing a selectivity of 8.6:1 towards the transport of Na₂SO₄ and NaCl. This value was higher than the Na₂SO₄/NaCl selectivity obtained with most of the polyamide membranes we tested in this study (5 out of 6; Figure 3D) and approached the highest selectivity achieved with NF270 polyamide membranes (10.7:1).

Comment 3 *Some important conclusions are lack of solid experimental evidence. For example, the authors mentioned "dehydration of the large divalent ions within the subnanometer pores", simulation or experimental evidences are required;*

Response We thank the reviewer for the suggestion to improve the comprehensiveness of this work. We have now added the molecular dynamics simulation to support our proposed pore-size-dependent dehydration-based selective ion transport. Briefly, we calculated the diffusion of Na⁺, Ca²⁺, Cl⁻, and SO₄²⁻ ions through aluminum oxide membranes with pore diameters of 10, 7.6, and 6 Å. We

noticed that all four ions retained their hydration shell within 10 Å pores. As a result, the selectivity between monovalent and divalent ions was close to 1 ($\text{Na}^+/\text{Ca}^{2+} = 1.0$ and $\text{Cl}^-/\text{SO}_4^{2-} = 1.1$). When the pore diameter was 7.6 Å, divalent ions started to lose their surrounding H_2O molecules while monovalent ions maintained their hydration state, leading to an enhanced selectivity between them ($\text{Na}^+/\text{Ca}^{2+} = 4.7$ and $\text{Cl}^-/\text{SO}_4^{2-} = 1.9$). When the pore diameter was 6 Å, all four ions were enforced to adjust their hydration shell, resulting in a lower selectivity ($\text{Na}^+/\text{Ca}^{2+} = 1.8$ and $\text{Cl}^-/\text{SO}_4^{2-} = 1.3$). These simulated results indeed verify the dominance of the hydration/dehydration state in differentiating the transport of monovalent and divalent ions.

The detail of the simulation was added to the Methods:

Lines 431-449: “Molecular Dynamics Simulations. The simulation box consisted of three main components: an amorphous Al_2O_3 membrane with subnanometer pores, a salt solution containing H_2O , Na^+ , Ca^{2+} , Cl^- , and SO_4^{2-} ions, and pure H_2O . To create the amorphous Al_2O_3 , a sample of crystalline Al_2O_3 (downloaded from the Materials Project) was heated at 3000 K for 100 ps using Materials Studio in Forcite Module with COMPASSII as the force field. The amorphous Al_2O_3 was then generated using the *supercell* function. The salt solution and pure H_2O were constructed using Amorphous Cell and combined with the amorphous Al_2O_3 using the *build* function.

The molecular structure of the simulation box was optimized using the Forcite Module with COMPASSII as the force field. During the geometric optimization, the convergence threshold for maximum energy change, maximum force, and maximum displacement were set to 0.001 kcal mol⁻¹, 0.5 kcal mol⁻¹ Å⁻¹, and 0.015 Å, respectively. To release the internal stress in the system, an MD simulation was performed under the NPT ensemble for 100 ps at 0.0001 GPa and 298 K until the density was stable over time. Further optimization of the system involved an MD simulation under the NVT ensemble for 100 ps. The simulation of ion passage involved the calculation of electrostatic interaction using Ewald and van der Waals force using Atom base. Precise Nose-Hoover temperature control mode and Berendsen pressure control mode were used in the simulation, and the concentration of ions permeating the membranes was calculated at the end of the MD simulation. All the simulations were conducted at the temperature of 298 K with a step size of 0.5 fs.”

The simulation results were added to the manuscript:

Figure 3. (F) Schematic of the simulation box consisting of a salt solution with 1.0 M of Na^+ , Ca^{2+} , Cl^- , and SO_4^{2-} ions (left), an AlO_x ALD membrane (middle), and pure H_2O (right). (G) Cross-sectional view of an AlO_x ALD membrane with 7.6 \AA pores. Five pores were drilled such that significant ion flux can be obtained in a concentration-gradient-driven process. (H) Simulated monovalent and divalent selectivity ($\text{Na}^+/\text{Ca}^{2+}$ and $\text{Cl}^-/\text{SO}_4^{2-}$) through AlO_x membranes with pore diameters of 6, 7.6, and 10 \AA .

Supplementary Figure 10. Hydration state of representative Na⁺, Ca²⁺, Cl⁻, and SO₄²⁻ ions in the bulk solution and inside the pores with diameters of 10, 7.6, and 6 Å.

We also added a relevant discussion in the main text:

Lines 297-313: “Molecular dynamics (MD) simulations were then performed to verify the dominant role of ion hydration/dehydration in differentiating the transport of monovalent and divalent ions. Salt solutions containing Na⁺, Ca²⁺, Cl⁻, and SO₄²⁻ ions and pure H₂O were placed on two sides of the AlO_x membrane (Fig. 3F). Five pores with identical diameters were drilled to facilitate the ion flux in a concentration-gradient-driven process (Fig. 3G). The selectivity between monovalent and divalent ions was calculated for AlO_x pores with diameters of 10, 7.6, and 6 Å to study the size-dependent ion selectivity.

When the pore diameter was 10 Å, all four ions (Na⁺, Ca²⁺, Cl⁻, and SO₄²⁻) retained a similar hydration shell within the pores compared to their respective hydration shell in the bulk solution (Supplementary Fig. 10). Consequently, the selectivity between monovalent and divalent ions was close to 1 (Na⁺/Ca²⁺ = 1.0 and Cl⁻/SO₄²⁻ = 1.1; Fig. 3H). When the pore diameter was narrowed to 7.6 Å, divalent ions partially lost surrounding H₂O molecules during their passage through the membrane pores while monovalent ions still maintained their original hydration shell (Supplementary Fig. 10), leading to an increased

monovalent/divalent selectivity ($\text{Na}^+/\text{Ca}^{2+} = 4.7$ and $\text{Cl}^-/\text{SO}_4^{2-} = 1.9$; Fig. 3H). Further narrowing the pore diameter to 6 Å enforced both the monovalent and divalent ions to adjust their hydration shell (Supplementary Fig. 10), resulting in a lower selectivity between them ($\text{Na}^+/\text{Ca}^{2+} = 1.8$ and $\text{Cl}^-/\text{SO}_4^{2-} = 1.3$; Fig. 3H).”

Comment 4 The membranes were not sufficiently characterized. For instance, the pore size and pore size distribution, porosities, and the surface hydrophilicity were unknown although they played significant roles in determining the separation performances. The calcined membranes took a grey color, implying there should be a considerable amount of carbon in the resulted porous layers; and the layers are expected to show somewhat a hydrophobic surface, which will hinder water permeance;

Response We thank the reviewer for the suggestion to improve the comprehensiveness of this work. The reviewer raised an interesting question about the characterization of membrane pore structure. Previous studies have used techniques, mainly nitrogen adsorption/desorption isotherms, to measure the pore size distribution and pore density within the newly developed membranes with subnanometer pores^{5,6}. However, it is now becoming evident that the nitrogen adsorption strategy cannot provide a satisfactory assessment of the distribution of micropores (pores < 2 nm)⁷. This is due to the difficulty of reaching the equilibrium at low pressure when measuring the adsorption isotherms and due to the blocking of narrow micropores with the pre-adsorbed nitrogen molecules⁷. Meanwhile, we also believe that the pore structure information (pore size distribution and pore density) obtained from the nitrogen adsorption analysis cannot be directly related to the transport performance of fabricated membranes. One reason is that pores measured using the nitrogen adsorption method can be non-continuous (i.e., not connecting the feed and permeate) and thus will not contribute to the transport properties. Another reason is that the separation capability of transport pathways is mainly governed by the narrowest regions within each pathway which can hardly be figured with the nitrogen adsorption technique.

Therefore, we decided to characterize the size of the membrane transport pathway by measuring the neutral solute transport in this work (Figure 4D and 4E). Characterizing membrane pore size based on the transport of neutral solutes has also been widely used in previous studies^{8,9}.

We have included our measured pore size in the main text:

Lines 340-343: “The observed solute flux suggests that the average pore **diameter** of the membranes fabricated with CH_3OH is between **6.6** and **8.1** Å, consistent with the average pore **diameter** estimated based on the selective transport between monovalent and divalent ions (~7.6 Å).”

Lines 348-350: “We note that negligible permeation of PEG 1000 was observed with the membranes created with $\text{CH}_3\text{CH}_2\text{OH}$, indicating that the average pore **diameter** of these membranes is between **10.1** and **15.5** Å.”

Lines 355-356: “This neutral solute transport measurement suggests that the average pore size of the AlO_x membranes constructed with $(\text{CH}_3)_2\text{CHOH}$ is between **15.7** and **23.0** Å.”

The water contact angle of the aluminum oxide membranes before and after calcination was measured based on the reviewer’s suggestion. Before calcination, a slight increase in the contact angle was observed when we increased the molecular weight of the alcohol used during the deposition (66.3, 72.6, and 75.3 ° for aluminum oxide films created with methanol, ethanol, and isopropanol, respectively). After burning off the organic decorations within the deposited hybrid films, all these membranes became highly hydrophilic with a contact angle of ~ 11 °. The contact

angle dropped quickly to 0 ° after the initial contact with the membrane surface. This is consistent with our observation that when the calcination temperature was over 600 °C, there was nearly no carbon residue within the hybrid films (Figure 2E and Supplementary Figure 7). To avoid potential confusion, we replaced the photo of the aluminum oxide membranes with the contact angle information:

Supplementary Figure 6. (A) SEM images of the MeO-AIO_x ALD film on top of the AAO substrates. Scale bars are 200 nm. (B) The water contact angle of the AIO_x membranes before (pink) and after (blue) calcination.

We also adjusted the discussion in the main text accordingly:

Lines 191-193: “No visible destruction of the ALD layer was observed after being calcinated for 5 h under an air atmosphere (Fig. 2D), and the membranes became hydrophilic (Supplementary Fig. 6B).

Comment The authors spent many words to discuss the advantage of ceramics. However, their membranes synthesized in this work are not ceramics; they were just amorphous aluminum oxides with some organic impurities;

5
Response We must acknowledge that we might not fully understand the reviewer’s comment, but we will try our best to improve the clarity of this manuscript. First of all, based on the XPS (Figure 2 and Supplementary Figure 7) and FTIR (Figure 2F) characterization, nearly no carbon residue was observed within the deposited films after the calcination. Therefore, the membranes synthesized in this work were mainly amorphous aluminum oxide membranes.

Ceramic membranes are defined as a type of artificial membranes made from inorganic materials, and they can be made from both crystalline and amorphous solids¹⁰. Based on this definition, the amorphous aluminum oxide membranes we fabricated can be classified as ceramic membranes. In fact, “ceramic” has been used to describe the metal oxide films deposited by the ALD technique. Here we listed a few examples:

- Atomic layer deposited conformal ceramic coatings for anti-corrosion of Ag nanoparticles. *Appl Surf Sci* 532, (2020).
- Synthesis of a novel porous polymer/ceramic composite material by low-temperature atomic layer deposition. *Chem Mater* 19, 5388-5394 (2007).
- Development of a thin ceramic-graphene nanolaminate coating for corrosion protection of stainless steel. *Corros Sci* 105, 161-169 (2016).
- Novel processing to produce polymer/ceramic nanocomposites by atomic layer deposition. *J Am Ceram Soc* 90, 57-63 (2007).

Comment 6 *The separation tests were performed in a concentration-diffusion mode. From the standpoint of practical applications, one would like to see the separation performances operated under pressures;*

Response We thank the reviewer for this suggestion and agree that further investigating the separation performance of the fabricated membranes in a pressure-driven process would be an interesting topic and can better represent the practical applications. However, the primary goal of this study is to propose a new strategy to fabricate aluminum oxide membranes with subnanometer pores and demonstrate the outstanding and tunable solute-solute separation capability of the fabricated membranes. Concentration-driven diffusion process has been widely used to characterize the solute-solute separation of newly developed membranes and thus should suffice the objective of our work. Here we listed a few examples:

- Precise and ultrafast molecular sieving through graphene oxide membranes. *Science* 343, 752-754 (2014).
- Ultrathin water-stable metal-organic framework membranes for ion separation. *Sci Adv* 6, (2020).
- Hydrophilic microporous membranes for selective ion separation and flow-battery energy storage. *Nat Mater* 19, 195-202 (2020).
- A lamellar MXene (Ti₃C₂T_x)/PSS composite membrane for fast and selective lithium-ion separation. *Angew Chem Int Edit* 60, 22265-22269 (2021).

The practical applications of our developed membranes, though important, is out of the scope of this work and can be investigated in our future studies.

Comment 7 *The authors are over exaggerated when commenting their results. For example, "for the first time the application of ceramic membranes in precise ion separations" in the Discussion part. Do the authors really have no aware of previous report on the application of ceramic membranes in ion separations?*

Response We thank the reviewer for this opportunity to improve the accuracy of our manuscript. The reviewer is correct that researchers have explored the application of ceramic membranes for ion separation before. However, despite the demonstration of ion separation, several facts need to be addressed. For instance, many studies relied on using composite materials as active layers for ion separation. Table R1 listed a few examples:

Table R1

Active layer	Supporting layer	Performance	Reference
UiO-66	Mullite hollow fiber	R _{NaCl} > 99%	11
ML-UiO-66	ZrO ₂ hollow fiber	R _{NaCl} > 99%	12
Crosslinked GOF	Al ₂ O ₃ hollow fiber	R _{NaCl} = 96%	13
Polyamide	Mullite hollow fiber	R _{NaCl} = 98%	14

When conventional ceramic membranes (i.e., metal oxide as selective layers) were designed for ion separation, researchers normally took advantage of electrostatic repulsion and focused on the rejection of multivalent co-ions or the separation between multivalent and monovalent co-ions. The pore diameters of their constructed membranes were usually above one nanometer, and thus steric confinement played a negligible role in impacting the ion transport and selectivity. One apparent limitation of these studies (or the electrostatic separation mechanism) is that both the membrane retention to multivalent co-ions and their selectivity towards the transport of multivalent and monovalent co-ions diminish with the enhanced feed solution concentration. Additionally, these

constructed membranes can barely reject multivalent counter-ions, and the membrane selectivity over the transport of multivalent and monovalent counter-ions is limited at the high feed concentration as well. Table R2 listed some examples:

Table R2

Active layer	Performance	Pore diameter	Reference
Al ₂ O ₃	10 mM feed: R _{CaCl₂} = 65%, R _{NaCl} = 10%, and R _{Na₂SO₄} = 0% 100 mM feed: R _{CaCl₂} = 12%, R _{NaCl} = 4%, and R _{Na₂SO₄} = 0%	8.8 nm	15
Al ₂ O ₃	10 mM feed: R _{CaCl₂} = 90%, R _{NaCl} = 55%, and R _{Na₂SO₄} = 0% 100 mM feed: R _{CaCl₂} = 40%, R _{NaCl} = 20%, and R _{Na₂SO₄} = 0%	1.6 nm	16

Therefore, implementing the subnanometer confinement should benefit the application of ceramic membranes for precise ion separation. For instance, our aluminum oxide membranes fabricated with methanol showed a selectivity of ~10:1 over the transport between NaCl and Na₂SO₄ and transport between NaCl and CaCl₂, at the concentration of 100 mM. To the best of our knowledge, our work pioneered the construction of ceramic membranes with subnanometer pores and demonstrated precise ion separation with extreme nanoconfinement. Nevertheless, in case we missed any significant literature, we deleted “for the first time”. **We also edited the sentence to make it more accurate:**

Lines 363-365: “This molecular-level design of the pore structure enables the application of ceramic membranes in precise ion separations **with subnanometer confinement.**”

Reviewer #2

General Comment *In this work, authors developed an ALD-based process to fabricate ceramic membranes with tunable subnanometer pores and demonstrated its use for precise ion separation.; excellent separation of NaCl from Na₂SO₄ has been shown, resulting from enforced dehydration of SO₄²⁻ and thus slower transport of Na₂SO₄. Adjustment of ALD parameters can be used to further tune the membrane pores for precise molecular separation. This is a novel work and is expected to have significant impact on ceramic membranes for precise separation. I would recommend publication after major revision for addressing the following questions and comments:*

Response We appreciate the reviewer for dedicating his/her/their time and effort to thoroughly review our manuscript. We are encouraged by the reviewer’s compliment on the novelty and significant impact of our study and hope to address the reviewer’s concerns in our detailed responses presented below.

Comment 1 *Ceramic membranes are more expensive compared to traditional polymeric membranes. However, one major advantage of ceramic membranes is that it can be used for applications where polymeric membranes would fail. How is using ceramic membranes justified for an application involving water where polymeric membranes can be readily used?*

Response We thank the reviewer for the opportunity of justifying using ceramic membranes over polymeric membranes for applications involving water. The reviewer is correct that polymeric membranes are commonly used during aqueous separations due to their low cost and outstanding separation performance. However, polymeric membranes are inapplicable in lots of harsh conditions. For instance, membrane technology has been employed to treat textile wastewater, and most textile

effluents have both high-temperature (up to 90 °C¹⁷) and high-basic nature (pH 11 or higher¹⁸). Using polymeric membranes to process the textile effluents significantly shortens their lifetime (e.g., the maximum operating temperature for polyamide thin-film composite membranes is 45 °C¹⁹). In these cases, ceramic membranes are preferred considering their tolerance to high temperatures and harsh chemicals. Furthermore, effective chemical cleaning can be conducted to control membrane fouling without the risk of membrane damage when using ceramic membranes. As a comparison, polymeric membranes (e.g., polyamide membranes) are vulnerable to chlorine attack, limiting the application of chlorine cleaning for membrane biofouling control. We have included these comparisons in the manuscript:

Lines 59-64: “Despite the versatility of the fabrication strategy and the tunability of the membrane chemistry, one major concern regarding employing polymeric membranes for selective separation is their instability under harsh operating conditions (e.g., high or low pH, presence of oxidants, and high temperatures). For instance, polyamide thin-film composite membranes, the state-of-the-art reverse osmosis membranes, are vulnerable to chlorine attack, substantially compromising their separation performance^{20, 21, 22}.”

Comment 2 *How do you prove that there is uniformity in distribution of molecular scale sacrificial fragments? The authors claim that the uniformity is necessary to eliminate pore collapse. And they ensure that there is uniformity of molecular scale sacrificial elements by using a low vapor pressure precursor (i.e., methanol). How do they prove this uniformity? Maybe a pore size distribution measurement after calcination would show a sharp, possibly more uniform sized pores. This can be used against a case where there is condensation (possibly carrying out the modified ALD reaction at low temperature to allow condensation), and the pore size distribution is wider.*

Response We thank the reviewer for helping improve the accuracy of our manuscript. After reconsidering the objective and novelty of this project, we believe that the uniformity in the distribution of sacrificial fragments is not the focus of the current work, and using the word “uniform” will confuse the readers. Here provides our detailed explanations.

In previous studies where porous membranes were constructed through calcinating the hybrid films prepared using MLD, low-volatility bifunctional organic precursors (e.g., EG) have to be used, leading to the condensation of organic molecules and the creation of large defects after the calcination⁴. To overcome this limitation, our objective is to develop a new deposition process that only involves highly volatile organic precursors and thus can create hybrid films with dispersed sacrificial fragments. The outstanding selectivity between monovalent and divalent ions obtained with our fabricated membranes (Figure 3C) confirms that the condensation of organic precursors and the resulting large pores can be avoided with our proposed monofunctional-alcohol-modulated ALD process as large pores deteriorate the membrane selectivity (Figure 3D). How uniformly these sacrificial fragments are distributed mainly impacts whether or to what extent these created pores are connected, which is out of the scope of the discussion in this work and can be studied in future work. Therefore, to improve the accuracy of this manuscript, we removed the word “uniform” from our manuscript (only appeared in one sentence in the penultimate paragraph of the Introduction):

Lines 83-86: “Hence, to realize the application of this sacrificial template strategy in constructing **defect-free** ceramic membranes with subnanometer pores, it is imperative to build hybrid films with distributed molecular-scale sacrificial segments.”

Although uniformity is not the focus of this work, we would like to still try our best to answer how uniformly these sacrificial fragments are distributed within the hybrid films since the reviewer raised this interesting question. Figure 1F indicates that the film growth rate is dependent on the

density of organic decorations; the higher the density is, the lower the growth rate will be. Therefore, one strategy for examining the uniformity of the organic decorations within the deposited films (or even within the deposition through the entire ALD chamber) is checking the film growth rate. We conducted the aluminum oxide deposition with the modulation of isopropanol and measured the growth rate at the inlet, the middle (3 different spots), and the outlet of the ALD chamber (Figure R1). The growth rates measured in the middle of the ALD chamber were all $0.23 \text{ \AA cycle}^{-1}$, the same as the growth rates measured at the inlet and outlet of the ALD chamber (both were $0.23 \text{ \AA cycle}^{-1}$). The consistency of the film growth rate through the entire ALD chamber, therefore, corroborates the uniformity of the organic decorations within the deposited films at the macroscale.

Figure R1. Photo of the ALD holder

In terms of the uniformity of the organic decorations at the atomic scale, although the reviewer suggested a conceptually feasible strategy (i.e., comparing the pore size distribution within the fabricated membranes at different deposition temperatures), we are afraid that this strategy will drastically change the deposition process in undesired ways and thus cannot provide the reliable evidence. Lowering the reactor temperature to where highly volatile monofunctional alcohols condense will lead to the condensation of H_2O as well, resulting in the chemical vapor deposition (instead of ALD) of aluminum oxide when we dose TMA. Additionally, the H_2O -based ALD process is much slower at lower temperatures, leaving the unreacted Al-CH_3 in the deposited films. Both factors will cause unpredictable changes to the microstructure of fabricated membranes.

Comment 3 *In MLD and ALD processes, usually the organic precursor is so chosen that after it reacts with the metal precursor, it ends up having one reactive end exposed (e.g., hydroxyl, amine etc.) This is done to ensure enough reactive sites for the next round of metal precursor. In this work, methanol was used along with water. The use of methanol limits the possibility of exposure to reactive groups in the next round of metallic precursor deposition. However, there is water that ensures reactive groups for the next cycle. Is the Methanol purged first followed by Water? Does the water react with unreacted Al-CH_3 groups, or it replaces Al-O-CH_3 groups? Why is some unreacted Al-CH_3 group remains after methanol purge? And if Water replaces Al-O-CH_3 groups, why doesn't it replace all of them? Also, mixing up two reactant (one with terminating group and another with reacting group) increases the chance of non-uniformity where there might be regions of carbon containing groups in the hybrid material and regions containing no hybrid groups. This may cause formation of pores that are not connected or are non-continuous, connecting the feed and permeating side.*

Response We thank the reviewer for this opportunity to clarify our deposition strategy. To simultaneously incorporate $-\text{OCH}_3$ groups and regenerate active $-\text{OH}$ groups on the substrate surface, a mixture of CH_3OH and H_2O was used as the co-reactants in our proposed ALD process. In practice, after purging the unreacted TMA precursors out of the reaction chamber, CH_3OH and H_2O were pulsed sequentially into the reaction chamber without break (i.e., purge step) between them. After reacting with the $-\text{CH}_3$ groups on the substrate surface, both the unreacted CH_3OH and H_2O were purged together out of the reaction chamber. We have introduced this procedure in the manuscript:

Lines 111-113: “After purging the reaction chamber with argon gas, a mixture of CH₃OH and H₂O was dosed sequentially without a purge break, with both co-reactants reacting with the –CH₃ groups in the following second half-cycle.”

According to Figure 1C, H₂O can replace –OCH₃ groups; based on Figure 1D, H₂O can directly react with unreacted –CH₃ groups as well. In practice, since we pulsed CH₃OH and H₂O together (sequentially without a purge break), we do not need to worry that there are no unreacted –CH₃ groups left after the CH₃OH purge, as mentioned by the reviewer. Also, it is more than likely that in our system H₂O will both replace –OCH₃ groups and directly react with unreacted –CH₃ groups.

The reviewer also raised an interesting question that if H₂O replaces –OCH₃ groups, why doesn't it replace all of them. We believe this is relevant to the novelty of our proposed ALD process. In a conventional ALD process, co-reactants (mainly H₂O) are overdosed to guarantee the full replacement of terminal groups on the substrate surface and to minimize the impurity of carbonaceous ligands within the deposited films. As a comparison, the objective of our proposed ALD process is to incorporate carbonaceous decorations within the deposited films which is conceptually different from the conventional ALD process. Hence, to achieve this objective, our strategy is overdosing the CH₃OH (e.g., 0.6 Torr) during the co-reactant pulse and minimizing the H₂O dosage (e.g., 0.007 Torr). In this way, the H₂O dosage is insufficient to replace all the –OCH₃ groups.

The reviewer is correct that mixing up two reactants will result in non-uniformity on the atomic scale since some sites are terminated with –OCH₃ groups and others are terminated with –OH groups. But on the nanometer scale and larger, the concentrations of these sites should be the same everywhere (partially supported by the measured film growth rate in the Response to Comment 2 from Reviewer 2). Therefore, the reviewer made a great comment on the existence of non-connected pores; in the current operation, we cannot rule out non-connected pores in these membranes. However, the observed non-zero water and solute permeance through the fabricated membranes indicates that at least some pores are connected. Moving forward, distributing the sacrificial segments uniformly at the atomic scale to further enhance the membrane permeability is an interesting research topic worth investigating, but is out of the scope of the discussion in this work.

Comment 4 *It is important to mention in the main text, the number of cycles needed to form a thin film on the AAO substrate.*

Response We thank the reviewer for this suggestion. We have already included the number of ALD cycles required to form a thin film on the AAO substrate in our main text.

Lines 177-179: “The minimum number of ALD cycles required to completely block the AAO pores was calculated based on the measured pore radius (~15.4 ± 0.7 nm, Supplementary Fig. 5) and the film growth rate (depending on the precursor dosage, Fig. 1F).”

Comment 5 *How was it ensured that the hybrid thin film formed is defect-free before organic removal and does not have any defects that are selective and provide ion selectivity?*

Response We thank the reviewer for this comment. To ensure that the hybrid thin film formed on top of the AAO substrate is defect-free before organic removal, we measured the water permeability of the ALD-deposited MeO-AlO_x hybrid films. According to Figure 3B, we can hardly observe any water flux before the calcination, suggesting that the deposited films are defect-free. As a comparison, after burning off the organic decorations by calcinating the hybrid film, water permeation was observed with the AlO_x membranes (Figure 3B).

Comment 6 *It is shown that changing the organic precursor has an effect on the tunability of the membrane; larger organic precursors make larger pores with higher permeance. Does the change of H₂O and methanol pulse pressure impart any tunability?*

Response We thank the reviewer for this question. The reviewer is correct that changing the H₂O or CH₃OH pulse pressure can also alter the transport properties of the fabricated membranes. For instance, when the pulse pressure of CH₃OH and H₂O was set to 0.4 and 0.06 Torr, respectively, we hardly observed any water flux through the fabricated membranes. As a comparison, when the pulse pressure of H₂O was decreased to 0.007 Torr, notably higher water permeance was observed for the fabricated membranes (2.6×10^{-3} LMH bar⁻¹; Figure 3B). We added this information to the manuscript.

Lines 233-235: **“No apparent water flux was observed for the calcinated AlO_x membranes created with 0.06 Torr of H₂O dosage. Therefore, the pulse pressure during membrane construction was set at 0.4 and 0.007 Torr for CH₃OH and H₂O dosage, respectively.”**

Comment 7 *The authors claim that a certain pulse pressure of methanol and water was promising to create continuous transport pathways. Why is this? What was the rationale behind choosing “0.4 and 0.007 Torr for CH₃OH and H₂O dosage, respectively”?*

Response We thank the reviewer for raising this important question and we do agree with the reviewer that claiming “the deposition condition with the respective pulse pressure of 0.4 and 0.007 Torr for CH₃OH and H₂O dosage is promising to create continuous transport pathways” based on “the calculation that each carbon atom is surrounded by ~6.1 oxygen atoms and ~2.9 aluminum atoms” is not rigorous. It is more appropriate to say that with the decreased H₂O pulse (0.007 Torr), it is more promising to create continuous transport pathways compared to the condition where 0.06 Torr of H₂O pulse was used. The rationale behind choosing “0.4 and 0.007 Torr for CH₃OH and H₂O dosage, respectively” should be the observed water flux with the membranes fabricated in these conditions. We adjusted the manuscript accordingly:

Lines 218-221: **“Specifically, for the MeO-AlO_x film deposited with 0.007 Torr of H₂O dosage, each carbon atom is surrounded by ~6.1 oxygen atoms and ~2.9 aluminum atoms (Supplementary Fig. 8), which is more promising for the creation of continuous transport pathways within the membrane compared to the film deposited with 0.06 Torr of H₂O dosage.”**

Lines 229-235: **“As shown in Fig. 3B, a notably higher water permeance was observed for the calcinated AlO_x membranes (2.6×10^{-3} LMH bar⁻¹) compared to the ALD deposited MeO-AlO_x films (both deposited with 0.007 Torr of H₂O dosage), suggesting that transport pathways can be created via burning off the decorated –OCH₃ groups. No apparent water flux was observed for the calcinated AlO_x membranes created with 0.06 Torr of H₂O dosage. Therefore, the pulse pressure during membrane construction was set at 0.4 and 0.007 Torr for CH₃OH and H₂O dosage, respectively.”**

Comment 8 *What was the temperature for calcination of the hybrid nanofilms? Was there any residual carbon in the film after calcination?*

Response We thank the reviewer for this question. The calcination temperature was set to 600 °C for the hybrid nanofilms. We mentioned this in the main text:

Lines 205-206: **“Hence, the calcination temperature was determined to be 600 °C for membrane fabrication.”**

Based on the XPS characterization (Figure 2E and Supplementary Figure 7), we noticed that when the calcination temperature was 600 °C or higher, nearly no signal was observed within the C 1s XPS spectrum, indicating that there was no residual carbon in the film after calcination.

Comment *The calcined hybrid film's surface charge should be measured. Because the AlO_x film in question*
9 *might have carbon residue and show some alteration of surface charge than that of AlO_x films*
reported in literature.

Response We thank the reviewer for this suggestion. The XPS measurement (Figure 2E and Supplementary Figure 7) suggests that there was no carbon residue within the AlO_x films after calcinating the hybrid films at 600 °C. Therefore, the AlO_x membranes we fabricated should have a similar surface charge as that of AlO_x reported in the literature. According to the literature, AlO_x is always positively charged at pH 5.7.^{23, 24, 25, 26, 27} Considering that electrostatic interaction was only used to interpret the faster permeation of SO₄²⁻ compared to Ca²⁺ ions which is not the main discussion of this work, we believe citing the positive charge of AlO_x from the literature should suffice.

Comment *Why was separation carried out at pH 5.7? is the pH for the real salt containing wastewater? Or it*
10 *was randomly chosen?*

Response We thank the reviewer for this comment. Our diffusion experiment was conducted in an open carbonate system, in which our aqueous solutions were in chemical equilibrium with atmospheric CO₂. Theoretically, an aqueous solution equilibrium with atmospheric CO₂ has a pH of 5.6.²⁸ Our measurement with the pH probe shows that the salt solutions we prepared have a pH of 5.7 which is consistent with the theoretical value.

Comment *The salt rejection experiments using the commercial membranes is to establish the role of pore size*
11 *in providing monovalent/divalent selectivity. While this is important to explain the*
monovalent/divalent selectivity, the experiment with commercial membrane is too distracting for
readers in the main text. This can be moved to the SI.

Response We thank the reviewer for this great suggestion. The reviewer is correct that providing too many details on the experiment of ion permeation through commercial PA TFC membranes is distracting for readers. On the other hand, this experiment is important to support the proposed hydration-state-based ion selectivity theory which is another novelty of this work and should be appealing to readers. Therefore, we have now moved the following experimental details to the Supplementary Information:

Supplementary Note 3: “The salt permeation rate through PA TFC membranes was measured with two typical RO membranes (SW30 and XLE with respective pore diameters of 5.7 and 5.9 Å; Supplementary Note 2, Supplementary Figs. 9A and 9B), a tight nanofiltration (NF) membrane (NF90 with a pore diameter of 6.8 Å; Supplementary Fig. 9C), a loose NF membrane (NF270 with the pore diameter of 7.5 Å; Supplementary Fig. 9D), and two tight UF membranes (UA60 and NDX with respective pore diameters of 9.7 and 10.4 Å; Supplementary Figs. 9E and 9F). The NaCl/Na₂SO₄ and NaCl/CaCl₂ selectivities for these membranes, obtained from salt permeation experiments using the diffusion cell, are summarized in Fig. 3D.”

We also adjusted the discussion in the main text accordingly:

Lines 283-285: “For ions permeating through pores similar to the size of the SO₄²⁻ ions, SO₄²⁻ ions have to adjust their hydration shell and thus need to overcome a high resistance during both pore entry and diffusion inside the pore.”

Comment *For the ions, diameter was used to explain the possible pore size of the membrane. It is suggested*
12 *that the authors also use diameter for neutral solutes (lines323-330).*

Response We thank the reviewer for the suggestion. We updated the main text and used diameter for the discussion of neutral solutes.

Lines 336-343: “For the AlO_x membranes created with CH_3OH (Figs. 4D and 4E), limited transport was observed when the solute **diameter** is larger than **8.1 Å** (sucrose of **8.2 Å** and raffinose of **10.0 Å**). When the solute **diameter** is smaller than **6.6 Å** (glucose of **6.5 Å**, erythritol of **5.3 Å**, and ethylene glycol of **4.0 Å**), these solutes can transport through the membrane pores, and their permeation rate increased with the decreasing solute **diameter**. The observed solute flux suggests that the average pore **diameter** of the membranes fabricated with CH_3OH is between **6.6** and **8.1 Å**, consistent with the average pore **diameter** estimated based on the selective transport between monovalent and divalent ions ($\sim 7.6 \text{ Å}$).”

Lines 348-360: “We note that negligible permeation of PEG 1000 was observed with the membranes created with $\text{CH}_3\text{CH}_2\text{OH}$, indicating that the average pore **diameter** of these membranes is between **10.1** and **15.5 Å**.”

Lines 355-356: “This neutral solute transport measurement suggests that the average pore size of the AlO_x membranes constructed with $(\text{CH}_3)_2\text{CHOH}$ is between **15.7** and **23.0 Å**.”

We also updated Figure 4:

Comment 13 *What was pulse pressure when other co-reactants (C2 and C3 alcohols) were used? Detailed methodology is essential. Also, it is mentioned that the calcination temperature was optimized; how was this optimized? What was the goal of the optimization? Least time? Or a certain pore size? Or something else. These details are missing.*

Response We thank the reviewer for pointing out these missing pieces of information. We added the pulse pressure for ethanol and isopropanol:

Lines 318-321: “Specifically, ethanol ($\text{CH}_3\text{CH}_2\text{OH}$; **0.2 Torr**) and isopropanol ($(\text{CH}_3)_2\text{CHOH}$; **0.2 Torr**) were employed to incorporate ethoxy ($-\text{OCH}_2\text{CH}_3$) and isopropoxy ($-\text{OCH}(\text{CH}_3)_2$) groups into the deposited films, respectively (Figs. 4A and 4B).”

The calcination temperature was optimized mainly to burn off the carbonaceous decorations to create membrane pores, as mentioned in the Methods:

Lines 398-400: “The ramp rate was maintained at 1°C min^{-1} and the calcinating temperature was optimized between 400 and 650 $^\circ\text{C}$ to burn off the carbonaceous species within the deposited film.”

Specifically, we used the XPS technique to quantify the carbon residue within hybrid films undergoing calcination at varied temperatures. We noticed that when the calcination temperature was 600 $^\circ\text{C}$ or higher, nearly no signal was observed within the C 1s XPS spectrum (Figure 2E and Supplementary Figure 7). Therefore, the calcination temperature was set at 600 $^\circ\text{C}$. We also included this information in the Results:

Lines 205-206: “Hence, the calcination temperature was determined to be 600 $^\circ\text{C}$ for membrane fabrication.”

Comment 14 Is it possible to prove experimentally or computationally the “possible dehydration of one ion to obtain selectivity” theory? It seems necessary to have some evidence for the ion dehydration claim. If it is indeed dehydration requirement for one ion to pass, which makes it difficult for it to pass through the pores, does that mean that the selectivity between the two ions observed under diffusion system would not be seen if the permeation was done under pressure (say at pressures higher to overcome this dehydration energy requirement? How much pressure would it be needed to eliminate the difficulty of dehydration?)

Response We thank the reviewer for the suggestion to improve the comprehensiveness of this work. Based on the reviewer’s comment, we added the molecular dynamics simulation to support our proposed pore-size-dependent dehydration-based selective ion transport. Briefly, we calculated the diffusion of Na^+ , Ca^{2+} , Cl^- , and SO_4^{2-} ions through aluminum oxide membranes with pore diameters of 10, 7.6, and 6 \AA . We noticed that all four ions retained their hydration shell within 10 \AA pores. As a result, the selectivity between monovalent and divalent ions was close to 1 ($\text{Na}^+/\text{Ca}^{2+} = 1.0$ and $\text{Cl}^-/\text{SO}_4^{2-} = 1.1$). When the pore diameter was 7.6 \AA , divalent ions started to lose their surrounding H_2O molecules while monovalent ions maintained their hydration state, leading to an enhanced selectivity between them ($\text{Na}^+/\text{Ca}^{2+} = 4.7$ and $\text{Cl}^-/\text{SO}_4^{2-} = 1.9$). When the pore diameter was 6 \AA , all four ions were enforced to adjust their hydration shell, resulting in a lower selectivity ($\text{Na}^+/\text{Ca}^{2+} = 1.8$ and $\text{Cl}^-/\text{SO}_4^{2-} = 1.3$). These simulated results indeed verify the dominance of the hydration/dehydration state in differentiating the transport of monovalent and divalent ions.

The detail of the simulation was added to the Methods:

Lines 431-449: “**Molecular Dynamics Simulations. The simulation box consisted of three main components: an amorphous Al_2O_3 membrane with subnanometer pores, a salt solution containing H_2O , Na^+ , Ca^{2+} , Cl^- , and SO_4^{2-} ions, and pure H_2O . To create the amorphous Al_2O_3 , a sample of crystalline Al_2O_3 (downloaded from the Materials Project) was heated at 3000 K for 100 ps using Materials Studio in Forcite Module with COMPASSII as the force field. The amorphous Al_2O_3 was then generated using the *supercell* function. The salt solution**

and pure H₂O were constructed using Amorphous Cell and combined with the amorphous Al₂O₃ using the *build* function.

The molecular structure of the simulation box was optimized using the Forcite Module with COMPASSII as the force field. During the geometric optimization, the convergence threshold for maximum energy change, maximum force, and maximum displacement were set to 0.001 kcal mol⁻¹, 0.5 kcal mol⁻¹ Å⁻¹, and 0.015 Å, respectively. To release the internal stress in the system, an MD simulation was performed under the NPT ensemble for 100 ps at 0.0001 GPa and 298 K until the density was stable over time. Further optimization of the system involved an MD simulation under the NVT ensemble for 100 ps. The simulation of ion passage involved the calculation of electrostatic interaction using Ewald and van der Waals force using Atom base. Precise Nose-Hoover temperature control mode and Berendsen pressure control mode were used in the simulation, and the concentration of ions permeating the membranes was calculated at the end of the MD simulation. All the simulations were conducted at the temperature of 298 K with a step size of 0.5 fs.”

The simulation results were added to the manuscript:

Figure 3. (F) Schematic of the simulation box consisting of a salt solution with 1.0 M of Na⁺, Ca²⁺, Cl⁻, and SO₄²⁻ ions (left), an AlO_x ALD membrane (middle), and pure H₂O (right). (G) Cross-sectional view of an AlO_x ALD membrane with 7.6 Å pores. Five pores were drilled such that significant ion flux can be obtained in a concentration-gradient-driven process. (H) Simulated monovalent and divalent selectivity (Na⁺/Ca²⁺ and Cl⁻/SO₄²⁻) through AlO_x membranes with pore diameters of 6, 7.6, and 10 Å.

Supplementary Figure 10. Hydration state of representative Na⁺, Ca²⁺, Cl⁻, and SO₄²⁻ ions in the bulk solution and inside the pores with diameters of 10, 7.6, and 6 Å.

We also added a relevant discussion in the main text:

Lines 297-313: “Molecular dynamics (MD) simulations were then performed to verify the dominant role of ion hydration/dehydration in differentiating the transport of monovalent and divalent ions. Salt solutions containing Na⁺, Ca²⁺, Cl⁻, and SO₄²⁻ ions and pure H₂O were placed on two sides of the AlO_x membrane (Fig. 3F). Five pores with identical diameters were drilled to facilitate the ion flux in a concentration-gradient-driven process (Fig. 3G). The selectivity between monovalent and divalent ions was calculated for AlO_x pores with diameters of 10, 7.6, and 6 Å to study the size-dependent ion selectivity.

When the pore diameter was 10 Å, all four ions (Na⁺, Ca²⁺, Cl⁻, and SO₄²⁻) retained a similar hydration shell within the pores compared to their respective hydration shell in the bulk solution (Supplementary Fig. 10). Consequently, the selectivity between monovalent and divalent ions was close to 1 (Na⁺/Ca²⁺ = 1.0 and Cl⁻/SO₄²⁻ = 1.1; Fig. 3H). When the pore diameter was narrowed to 7.6 Å, divalent ions partially lost surrounding H₂O molecules during their passage through the membrane pores while monovalent ions still maintained their original hydration shell (Supplementary Fig. 10), leading to an increased

monovalent/divalent selectivity ($\text{Na}^+/\text{Ca}^{2+} = 4.7$ and $\text{Cl}^-/\text{SO}_4^{2-} = 1.9$; Fig. 3H). Further narrowing the pore diameter to 6 Å enforced both the monovalent and divalent ions to adjust their hydration shell (Supplementary Fig. 10), resulting in a lower selectivity between them ($\text{Na}^+/\text{Ca}^{2+} = 1.8$ and $\text{Cl}^-/\text{SO}_4^{2-} = 1.3$; Fig. 3H).”

The reviewer also raised an interesting question on the impact of the applied hydraulic pressure on ion dehydration and selective ion transport. According to previous studies, even in a pressure-driven nanofiltration process, ions still need to undergo dehydration to enter the membrane pores, and the different ease of dehydration would result in the selective ion transport.²⁹

Reviewer #3

General Comment *This paper reports the preparation of subnanometer porous alumina selective layer supported on AAO substrate by using the monofunctional-alcohol-modulated atomic layer deposition method. The subnanometer pores are created by burning off organic moieties. This method is elegant and novel and provides an effective way to accurately control the pore size of ceramic membrane. The membranes are well characterized and show high monovalent to divalent ion selectivity. The manuscript is well written and can be accepted for publication after some minor revision.*

Response We truly thank the reviewer for this very positive and supporting summary of our work. We are extremely encouraged by the reviewer’s compliment on the comprehensiveness and novelty of our study.

Comment 1 *After the ALD process, the membranes are calcined in air to generate pores. It would be better to perform control experiments that the membranes are calcined in inert gas such as argon at the same temperatures to see how the membrane structure changes. This will help understand the pore formation mechanisms.*

Response We thank the reviewer for the suggestion. As per their recommendation, we conducted a thermogravimetric analysis (TGA) of the aluminum oxide membranes fabricated with isopropanol under environments of oxygen gas and argon gas. According to Figure R2, we observed that heating the membranes in an oxygen atmosphere caused a gradual decrease in their mass from 100.08% to 99.83% as the temperature increased from 50 to 600 °C. However, when heating the membranes in an argon atmosphere, their mass remained nearly constant (100.07% and 100.4% at 50 and 600 °C, respectively). These findings suggest that the presence of oxygen is necessary to burn off all the carbonous decorations and create porous structures.

Figure R2. TGA analysis of the aluminum oxide membranes fabricated with isopropanol under environments of oxygen gas and argon gas.

Reference:

1. Wu S, Wang Z, Xiong S, Wang Y. Tailoring TiO₂ membranes for nanofiltration and tight ultrafiltration by leveraging molecular layer deposition and crystallization. *Journal of Membrane Science* **578**, 149-155 (2019).
2. Chen H, Jia X, Wei M, Wang Y. Ceramic tubular nanofiltration membranes with tunable performances by atomic layer deposition and calcination. *Journal of Membrane Science* **528**, 95-102 (2017).
3. Song Z, *et al.* TiO₂ nanofiltration membranes prepared by molecular layer deposition for water purification. *Journal of Membrane Science* **510**, 72-78 (2016).
4. Liang XH, Yu M, Li JH, Jiang YB, Weimer AW. Ultra-thin microporous-mesoporous metal oxide films prepared by molecular layer deposition (MLD). *Chem Commun*, 7140-7142 (2009).
5. Li XY, *et al.* Fast and selective fluoride ion conduction in sub-1-nanometer metal-organic framework channels. *Nat Commun* **10**, (2019).
6. Fan HW, Peng MH, Strauss I, Mundstock A, Meng H, Caro J. MOF-in-COF molecular sieving membrane for selective hydrogen separation. *Nat Commun* **12**, (2021).
7. Thommes M, *et al.* Physisorption of gases, with special reference to the evaluation of surface area and pore size distribution (IUPAC Technical Report). *Pure Appl Chem* **87**, 1051-1069 (2015).
8. Nghiem LD, Schafer AI, Elimelech M. Removal of natural hormones by nanofiltration membranes: Measurement, modeling, and mechanisms. *Environmental Science & Technology* **38**, 1888-1896 (2004).
9. Xie M, Nghiem LD, Price WE, Elimelech M. Relating rejection of trace organic contaminants to membrane properties in forward osmosis: Measurements, modelling and implications. *Water Research* **49**, 265-274 (2014).
10. Ceramic membrane in Wikipedia. https://en.wikipedia.org/wiki/Ceramic_membrane.
11. Li HT, *et al.* Stable Zr-Based Metal-Organic Framework Nanoporous Membrane for Efficient Desalination of Hypersaline Water. *Environmental Science & Technology* **55**, 14917-14927 (2021).
12. Wang XL, *et al.* Robust ultrathin nanoporous MOF membrane with intra-crystalline defects for fast water transport. *Nat Commun* **13**, (2022).
13. Yuan BQ, *et al.* Cross-linked Graphene Oxide Framework Membranes with Robust Nano-Channels for Enhanced Sieving Ability. *Environmental Science & Technology* **54**, 15442-15453 (2020).
14. Zhang MM, Jin WB, Yang FL, Duke M, Dong YC, Tang CY. Engineering a Nanocomposite Interlayer for a Novel Ceramic-Based Forward Osmosis Membrane with Enhanced Performance. *Environmental Science & Technology* **54**, 7715-7724 (2020).

15. Qi H, Niu SF, Jiang XL, Xu NP. Enhanced performance of a macroporous ceramic support for nanofiltration by using alpha-Al₂O₃ with narrow size distribution. *Ceram Int* **39**, 2463-2471 (2013).
16. Chen XF, Zhang W, Lin YQ, Cai YY, Qiu MH, Fan YQ. Preparation of high-flux gamma-alumina nanofiltration membranes by using a modified sol-gel method. *Micropor Mesopor Mat* **214**, 195-203 (2015).
17. Agtas M, Yilmaz O, Dilaver M, Alp K, Koyuncu I. Hot water recovery and reuse in textile sector with pilot scale ceramic ultrafiltration/nanofiltration membrane system. *J Clean Prod* **256**, (2020).
18. Agtas M, Dilaver M, Koyuncu I. Ceramic membrane overview and applications in textile industry: a review. *Water Sci Technol* **84**, 1059-1078 (2021).
19. El-Arnaouty MB, Ghaffar AMA, Eid M, Aboufotouh ME, Taher NH, Soliman E. Nano-modification of polyamide thin film composite reverse osmosis membranes by radiation grafting. *J Radiat Res Appl Si* **11**, 204-216 (2018).
20. Do VT, Tang CYY, Reinhard M, Leckie JO. Effects of Chlorine Exposure Conditions on Physiochemical Properties and Performance of a Polyamide Membrane-Mechanisms and Implications. *Environmental Science & Technology* **46**, 13184-13192 (2012).
21. Shao FF, *et al.* Graphene oxide modified polyamide reverse osmosis membranes with enhanced chlorine resistance. *Journal of Membrane Science* **525**, 9-17 (2017).
22. Yao YJ, Zhang PX, Jiang C, DuChanois RM, Zhang X, Elimelech M. High performance polyester reverse osmosis desalination membrane with chlorine resistance. *Nat Sustain* **4**, 138-146 (2021).
23. Robinson M, Pask JA, Fuerstenau DW. Surface Charge of Alumina and Magnesia in Aqueous Media. *Journal of the American Ceramic Society* **47**, 516-520 (1964).
24. Goyne KW, Zimmerman AR, Newalkar BL, Komarneni S, Brantley SL, Chorover J. Surface charge of variable porosity Al₂O₃(s) and SiO₂(s) adsorbents. *J Porous Mat* **9**, 243-256 (2002).
25. Bahena JLR, Cabrera AR, Valdivieso AL, Urbina RH. Fluoride adsorption onto alpha-Al₂O₃ and its effect on the zeta potential at the alumina-aqueous electrolyte interface. *Sep Sci Technol* **37**, 1973-1987 (2002).
26. Wisniewska M, Chibowski S, Urban T. Comparison of adsorption affinity of ionic polyacrylamide for the surfaces of selected metal oxides. *Adsorpt Sci Technol* **35**, 582-591 (2017).
27. Xia ZJ, Rozyyev V, Mane AU, Elam JW, Darling SB. Surface Zeta Potential of ALD-Grown Metal-Oxide Films. *Langmuir* **37**, 11618-11624 (2021).
28. The Open Carbonate System. <https://www.aqion.de/site/161>.
29. Epsztein R, Shaulsky E, Dizge N, Warsinger DM, Elimelech M. Role of Ionic Charge Density in Donnan Exclusion of Monovalent Anions by Nanofiltration. *Environmental Science & Technology* **52**, 4108-4116 (2018).

Reviewers' Comments:

Reviewer #1:

Remarks to the Author:

The manuscript has been somewhat improved after revision; however, I am not convinced to accept it as its weaknesses in novelty, performances, and potential applications remained.

Reviewer #2:

Remarks to the Author:

The authors did a very good job responding to my comments and suggestions. The manuscript has been greatly improved by addressing potential concerns and adding additional experimental and simulation results. I would recommend acceptance of the revised manuscript.

Reviewer #3:

Remarks to the Author:

The authors have addressed the reviewers' comments well. The revised manuscript can be accepted for publication.

Response to Reviewers' Comments

In the following response letter, black italic type represents the exact comments from the reviewers, blue type represents our response, green type represents unmodified text in the manuscript, and bold green type represents new text added to the revised manuscript. Lines listed refer to the revised manuscript.

Reviewer #1

General Comment *The manuscript has been somewhat improved after revision; however, I am not convinced to accept it as its weaknesses in novelty, performances, and potential applications remained.*

Response We thank the reviewer for his/her/their time and effort in reviewing our manuscript. In our previous response to Reviewer 1, we addressed the reviewer's concern about the novelty of our work by clarifying the conceptual difference between our proposed monofunctional-alcohol-modulated atomic layer deposition process and the conventional deposition methods. We have also highlighted the importance of utilizing high-volatility coreactants to produce subnanometer pores with precise control, a feat that was previously unattainable with other methods.

In terms of membrane performances, we compared the selectivity toward the transport of divalent (Na_2SO_4) and monovalent (NaCl) salts of our methanol-based membranes and of six commercially available polyamide thin-film composite membranes. Our results indicate that our membranes outperformed five out of the six commercial membranes tested, approaching the highest selectivity observed with NF270 membranes.

Although there is still work to be done to fully realize the potential of these membranes in practical applications, we believe that this work represents a significant breakthrough in terms of demonstrating the feasibility of our fabrication strategy.

As Reviewer 3 noted, "*The authors have addressed the reviewers' comments well. The revised manuscript can be accepted for publication.*" We sincerely hope that Reviewer 1 will also support the publication of our work.

Reviewer #2

General Comment *The authors did a very good job responding to my comments and suggestions. The manuscript has been greatly improved by addressing potential concerns and adding additional experimental and simulation results. I would recommend acceptance of the revised manuscript.*

Response We truly thank the reviewer for constructive comments and suggestions, which helped us significantly improve the quality of our work.

Reviewer #3

General Comment *The authors have addressed the reviewers' comments well. The revised manuscript can be accepted for publication.*

Response We appreciate the reviewer for the enormous support of our work.